# Flat clathrin lattices are dynamic actin-controlled hubs for clathrin-mediated endocytosis and signalling of specific receptors

Daniela Leyton-Puig[1,*], Tadamoto Isogai[2,*,†], Elisabetta Argenzio[1], Bram van den Broek[1], Jeffrey Klarenbeek[1], Hans Janssen[3], Kees Jalink[1] & Metello Innocenti[2]

Clathrin lattices at the plasma membrane coat both invaginated and flat regions forming clathrin-coated pits and clathrin plaques, respectively. The function and regulation of clathrin-coated pits in endocytosis are well understood but clathrin plaques remain enigmatic nanodomains. Here we use super-resolution microscopy, molecular genetics and cell biology to show that clathrin plaques contain the machinery for clathrin-mediated endocytosis and cell adhesion, and associate with both clathrin-coated pits and filamentous actin. We also find that actin polymerization promoted by N-WASP through the Arp2/3 complex is crucial for the regulation of plaques but not pits. Clathrin plaques oppose cell migration and undergo actin- and N-WASP-dependent disassembly upon activation of LPA receptor 1, but not EGF receptor. Most importantly, plaque disassembly correlates with the endocytosis of LPA receptor 1 and down-modulation of AKT activity. Thus, clathrin plaques serve as dynamic actin-controlled hubs for clathrin-mediated endocytosis and signalling that exhibit receptor specificity.

[1] Division of Cell Biology I, The Netherlands Cancer Institute, Plesmanlaan 121, Amsterdam 1066 CX, The Netherlands. [2] Division of Molecular Genetics, The Netherlands Cancer Institute, Plesmanlaan 121, Amsterdam 1066 CX, The Netherlands. [3] Division of Cell Biology II, The Netherlands Cancer Institute, Plesmanlaan 121, Amsterdam 1066 CX, The Netherlands. * These authors contributed equally to this work. † Present address: Department of Cell Biology; Lyda Hill Department of Bioinformatics, University of Texas Southwestern Medical Center, 6000 Harry Hines Blvd, Dallas, Texas 75390, USA. Correspondence and requests for materials should be addressed to K.J. (email: k.jalink@nki.nl) or to M.I. (email: m.innocenti@nki.nl).

Cells internalize membrane proteins, solutes and lipids through the formation of clathrin-coated vesicles (CCVs), a process referred to as clathrin-mediated endocytosis (CME). CME entails five stages: initiation, cargo recruitment, clathrin coat assembly, scission of a CCV and subsequent uncoating thereof[1]. In a widely accepted model, binding of clathrin adaptor proteins to the plasma membrane recruits clathrin triskelia, thereby promoting the self-assembly of a clathrin coat that marks *de novo* endocytic sites. At these sites, clathrin-coated pits (CCPs, hereafter referred to as pits) mature, recruit cargoes and ultimately pinch off with the help of the GTPase dynamin to form small and roughly spherical CCVs of up to 200 nm in diameter[1,2].

Pharmacological studies have suggested that actin polymerization optimizes CME of epidermal growth factor receptor (EGFR) and some G-protein coupled receptors[3,4]. However, actin has a non-obligatory and cell-type-specific role in CME of Transferrin Receptor[5,6].

Knockdown studies showed that N-WASP and the Arp2/3 complex mediate the assembly of F-actin on CCPs and vesicles[7,8]. Consistently, live-cell experiments demonstrated that actin appears on pits just before scission and only after N-WASP and the Arp2/3 complex[6,9–12]. Thus, actin polymerization likely provides mechanical force for pit remodelling and scission[11,13,14]. In addition, recent data indicate that membrane tension may determine whether or not CME depends on actin[15]. As CME is the main route for membrane protein internalization[16], it is not surprising that it affects signalling of receptor tyrosine kinases and G-protein coupled receptors[1]. By removing activated receptors from the cell surface, CME can either attenuate or elicit the activity of specific downstream signalling pathways[17].

Electron microscopy (EM) and total internal reflection fluorescence (TIRF) microscopy showed that, in addition to the curved pits and CCVs, a second type of clathrin structures exists on the membrane of cells, namely large clathrin structures that are often referred to as flat clathrin plaques (FCPs, hereafter referred to as plaques)[11,18,19]. The characteristic geometry and curvature of pits and plaques arises from a different assemblage of clathrin triskelia[20]: a combination of pentagons and hexagons determines the basket-like shape and curvature of the coat surrounding CCVs, whereas hexagonal only honey-comb-like structures give rise to plaques[20].

The function of plaques is much debated: some studies concluded that plaques are endocytically inactive, long-lived structures[21,22], whereas other studies found that they can be actively internalized[13], or serve as focal sites of CCV formation[11,23,24]. At any rate, CCVs are often found to surround the borders of plaques in EM images[5,18,19,25]. In addition, plaques and long-lived clathrin-coated structures (CCSs) have been suggested to be sites of adhesion[13,21,25].

Light microscopy of clathrin tagged with, for example, green fluorescent protein (GFP) has been instrumental to illuminate the spatiotemporal mechanics of CME[26–29]. This approach has shown that convex pits and flat plaques exhibit distinctive persistence and brightness on the plasma membrane[13,14,22,23]. However, the diffraction-limited resolution of the light microscope has hampered more detailed morphometric analyses and makes the discrimination between pits and plaques challenging because of their small size.

We combine super-resolution (SR) microscopy, molecular genetics and cell biology to study in great detail the function and the regulation of plaques. Here, we report that plaques are dynamic structures associating with both actin filaments and the cell substrate and that they are sites of CCV formation. By depleting N-WASP and the Arp2/3 complex, and using dominant negative mutants of N-WASP, we show that actin polymerization controls plaque dynamics. Finally, we demonstrate that plaques are involved in cell migration and function as hubs for CME and signalling of the LPA1 receptor (LPAR1). In summary, these data shed light on the enigmatic function of plaques and unveil an actin-based mechanism regulating the lifecycle of these clathrin-coated nanodomains.

## Results

**Clathrin-coated structures by SR microscopy**. We used correlative TIRF and highly optimized GSDIM SR microscopy[30,31] (see also Methods) to characterize the CCSs on the plasma membrane of cells. We imaged endogenous clathrin heavy chain (CHC) at the basal membrane of HeLa cells, which assemble both pits and plaques[13,22].

Small round structures ($\sim$100–200 nm diameter, hereafter referred to as pits) in which the lumen could often distinctly be resolved, and larger heterogeneous structures (hereafter referred to as plaques) were visible in the SR images (Fig. 1a and Supplementary Fig. 1a,b for a gallery of representative objects and classification thereof). Notably, pits that are juxtaposed to or overlapping with plaques would be easily misclassified as plaques using conventional light microscopy (Fig. 1a,b). In contrast, the higher resolving power of SR allowed reliable segmentation and morphometric analyses of both categories (see also Methods). Small and circular pits measured $120 \pm 25$ nm in diameter whereas plaques were substantially larger and highly heterogeneous in shape (Fig. 1c), in line with earlier reports[13,22,25]. Furthermore, CHC and GFP-tagged clathrin light chain (CLC) colocalized in dual-colour SR images thereby ruling out bias by the employed antibodies (Supplementary Fig. 2a), and both pits and plaques could be detected in our HeLa cells also by EM (Supplementary Fig. 2b). Thus, this SR-based approach appears well suited to study CCSs.

To obtain a molecular characterization of pits and plaques, we assessed the presence of key components of the machinery regulating CME and adhesion to the extracellular matrix. SR clearly showed that CLC, adaptor protein complex 2 and Dynamin II localized both in pits and plaques (Supplementary Fig. 2a,c,d). These data are in line with previous live-cell TIRF studies identifying the same CME machinery at both short-lived and long-lived CCSs[12]. Two-colour SR images of HeLa cells showed in many cases a clear enrichment of F-actin around both CHC-positive and CLC-positive structures (Fig. 1d). This is consistent with EM studies in which actin was found to associate with both pits and plaques[12]. Interestingly, N-WASP, which activates the Arp2/3 complex and mediates actin-dependence of CME[7,8], was present in plaques (Fig. 1e). Using a coordinate-based colocalization quantification algorithm[32], we found N-WASP to be moderately enriched at these sites (Fig. 3d). This is consistent with the very transient and late recruitment of the actin machinery during CME[12] and also likely explains the occasional actin-negative plaques observed in Fig. 1d. In addition, the N-WASP-regulator and actin-binding protein Cortactin[33] was present in both pits and plaques (Supplementary Fig. 2e). Extracellular matrix-binding protein Integrin β5 localized to both pits and plaques, as previously reported[34,35] (Supplementary Fig. 2f).

In summary, SR shows that clathrin plaques are often juxtaposed to CCVs and contain both F-actin and the machinery mediating CCV formation and cell-matrix adhesion. Thus, plaques may be substrate-contact points and sites of actin-dependent formation of CCVs.

**N-WASP is a key regulator of plaques**. To explore this latter idea, we imaged the CCSs in control knockdown (KD) and N-WASP KD HeLa cells (Fig. 2a). Silencing of N-WASP not only resulted in CCSs devoid of F-actin as we previously showed[8]

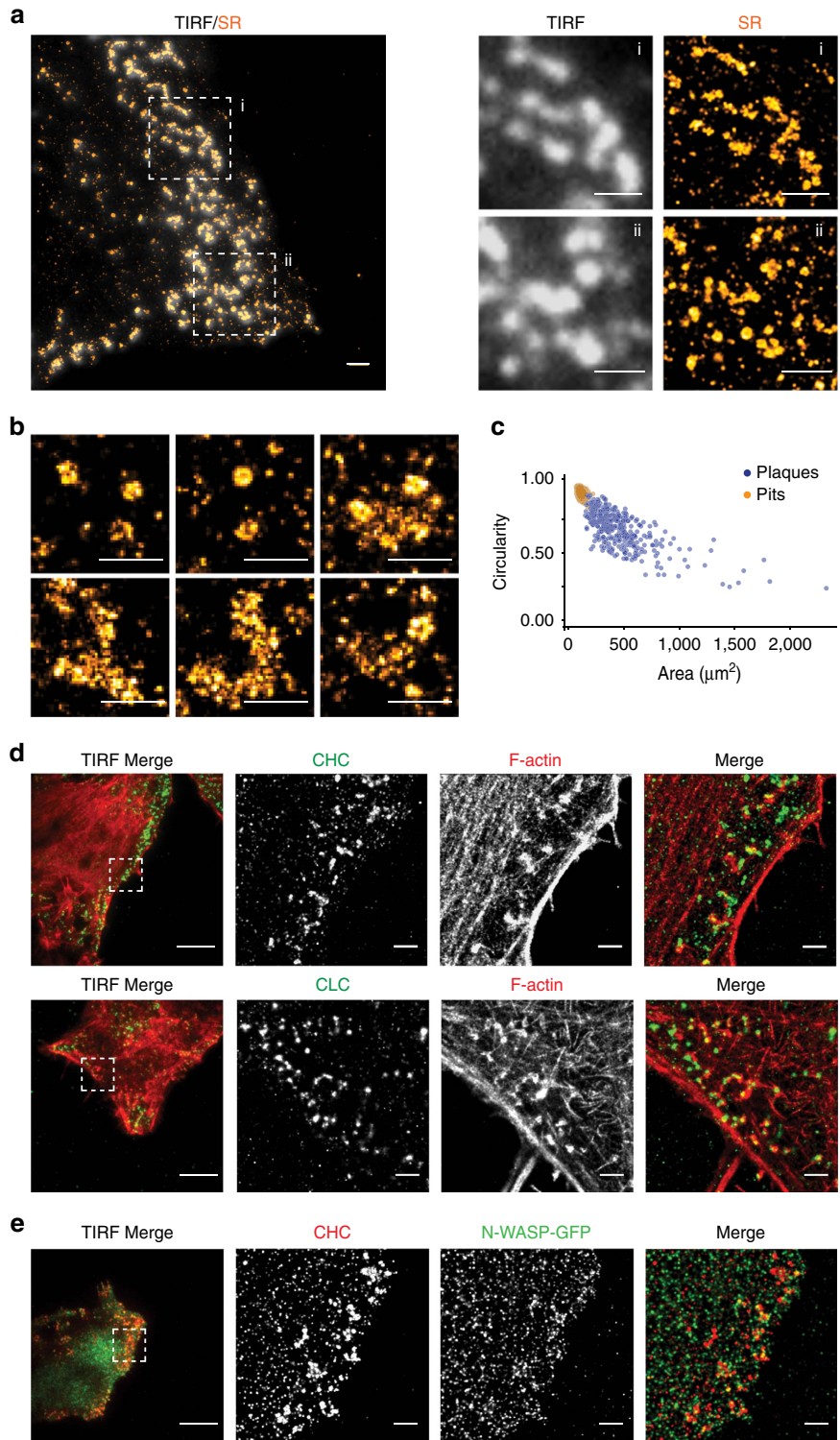

**Figure 1 | SR microscopy enables in-depth characterization of clathrin-coated structures.** (**a**) Representative correlative TIRF-SR image of the CCSs on the basal membrane of HeLa cells. HeLa cells were fixed and stained with anti-clathrin heavy chain (CHC) antibodies as indicated in the Methods. Overlay of the TIRF (grey) and the SR (orange) images and zoomed-in regions (i, ii) are depicted. Scale bar, 1 μm. (**b**) Gallery of SR images showing the diversity of the CCSs (pits, plaques and combination thereof) found on the basal membrane of HeLa cells. Scale bar, 1 μm. (**c**) Morphometric analysis of CCSs on the basal membrane of HeLa cells. Circularity (1 = high circularity, 0 = high asymmetry) and surface area (μm$^2$) of individual CCS were obtained as described in the Methods. (**d**) Actin associates with CCSs. Representative two-colour TIRF and SR images of F-actin (red in merge) with either CHC or CLC tagged with mTurquoise2 (CLC-mTQ2) (green in merge). HeLa cells were fixed and stained with anti-CHC or anti-GFP antibodies and phalloidin to detect CCSs and F-actin, respectively. CLC-mTQ2 was transfected as described in the Methods. Scale bar, 1 μm. (**e**) N-WASP associates with CCSs. Representative two-colour TIRF and SR images of CHC (red in merge) and N-WASP tagged with GFP (N-WASP-GFP, green in merge). HeLa cells were transfected, processed and imaged as illustrated above. Scale bar, 1 μm.

(Fig. 2b), but also in a significant broadening of the peripheral CCS-rich ring at the basal membrane, as visualized by TIRF imaging (Fig. 2c). Most importantly, SR images of the CCS-rich outer ring of the N-WASP KD cells showed that it was composed predominantly of plaques (Fig. 2c, Supplementary Fig. 3a,b). Morphometric analyses revealed that plaques covered a much larger area of the N-WASP KD cells as compared to the control cells (Fig. 2d). In contrast, the knockdown of N-WASP had no effect on the number of pits (Fig. 2d). Rescue of the N-WASP KD cells with shRNA-resistant N-WASP reduced the area covered by plaques without perturbing pits (Fig. 2e,f).

To test the importance of N-WASP in the control of plaques, we took advantage of BSC-1 cells, which reportedly do not form plaques and only contain pits[2,13]. Strikingly, upon silencing of N-WASP, BSC-1 cells presented large plaques (Supplementary Fig. 4), the size and shape of which were comparable with those found in HeLa cells. These results confirm that the involvement of N-WASP in plaque regulation is common to different cell types.

**N-WASP controls plaques by activating the Arp2/3 complex.** To elucidate how N-WASP regulates plaques, we exploited a series of N-WASP deletion mutants (Fig. 3a). N-WASP is a multi-domain protein containing a C-terminal VCA region that binds and activates the Arp2/3 complex[33]. In resting conditions, the VCA region establishes an intra-molecular interaction with the GTPase-binding domain (GBD) and the Basic (B) region that locks N-WASP in an auto-inhibited conformation[33]. Concomitant binding of activated Cdc42 to the GBD and either phosphatidylinositol 4,5-biphosphate ($PIP_2$) to the B region or SH3-containing proteins to the central Proline-rich domain (PRD) are needed to efficiently relieve N-WASP from auto-inhibition[33]. Instead, the N-terminal WASP homology domain 1 (WH1) mediates binding to WASP-interacting proteins, which stabilizes N-WASP and strengthens auto-inhibition[33].

N-WASP mutants that lack the VCA domain (ΔVCA) do not bind the Arp2/3 complex and consequently fail to induce actin polymerization. GFP-tagged N-WASP ΔVCA strongly localized to plaques in control KD HeLa cells and caused plaques to cover a much larger area of the membrane than in GFP-transfected cells (Fig. 3b–d). This is in line with the phenotype of the N-WASP KD cells and suggests the involvement of actin polymerization and the Arp2/3 complex in plaque regulation. Expression of the ΔWH1 and ΔPRD mutants exerted similar effects, even though they were not enriched in plaques (Fig. 3b–d). Hence, it is likely that these mutants deplete factors that are required for endogenous N-WASP to function at plaques. Interestingly, overexpression of full-length N-WASP caused a significant reduction of the area covered by plaques, whereas a functionally dead point mutant (H208D) that cannot bind Cdc42 (ref. 7) did not significantly localize or perturb plaques (Fig. 3b–d). The observation that neither N-WASP nor its mutants affected the number of pits (Fig. 3e) and N-WASP's localization to CCSs (Fig. 3d,f) jointly unmask a seemingly more prominent role for N-WASP in the regulation of plaques than in that of pits. Most importantly, our results suggest that N-WASP exploits Arp2/3-complex-dependent actin polymerization to control plaques.

To strengthen this notion, we knocked down the Arp2/3-complex core subunit Arpc2 (ref. 36) thereby causing downregulation of the entire Arp2/3 complex in HeLa cells (Fig. 3g). The silencing of the Arp2/3 complex phenocopied that of N-WASP in these cells: both knockdowns exhibited a bigger area covered by plaques but no change in the pit number (Fig. 3h,i). Furthermore, over-expression of full-length N-WASP in the Arpc2 KD cells did not decrease the area covered by plaques (Fig. 3j), thus showing that N-WASP controls plaques by activating the Arp2/3 complex.

Taken together, these results indicate that actin polymerization induced by N-WASP and the Arp2/3 complex has a key role in plaque regulation.

**Plaques remodel in response to external stimuli.** N-WASP responds to a variety of external cues that activate Cdc42 (ref. 33), including fetal calf serum (FCS, hereafter referred to as serum)[37]. Indeed, withdrawal of serum caused a significant increase in the plaque-covered area (Fig. 4a). We investigated whether serum acts through N-WASP by measuring the area covered by plaques in control and N-WASP KD cells that were kept in either high (10%) or low (0.1%, serum starvation) serum (Fig. 4b). In control cells, serum starvation caused a threefold increase in plaques. A much less dramatic effect ($\sim$1.5-fold) was seen in N-WASP KD cells, with the remaining increase most probably due to the residual N-WASP expression. To characterize how cells remodel plasma-membrane CCSs in response to serum, we took serum-starved control KD cells and imaged them before and after acute stimulation with serum for 3, 7, 15, 30 and 60 min. SR revealed that the area covered by plaques rapidly decreased, whereas the pit number did not vary (Fig. 4c). These data show that plaques are highly dynamic structures under the control of external cues. Live-cell TIRF microscopy and tracking of discrete CCSs in cells expressing RFP-tagged CLC confirmed this result, as CCS persistence after serum stimulation was $\sim$4-fold longer in N-WASP KD cells than in control ones (Fig. 4d and Supplementary Movie 1).

The bioactive lipid lysophosphatidic acid (LPA), a major serum component that affects various cellular functions[38] fully recapitulated the effects of serum (Fig. 4e,f). In contrast, epidermal growth factor (EGF) had no effect on plaques (Fig. 4e,f), even though it is found in serum[8]. Of note, neither LPA nor EGF affected the number of pits significantly. Most importantly, plaque dissolution is not a spurious effect of LPA as it was virtually suppressed by knocking down LPA receptor 1 (LPAR1) (Fig. 4g,h).

Taken together, these findings show that N-WASP-controlled plaques are dynamic structures that respond to specific extracellular stimuli.

**Plaques are sites of CCV formation.** To study the dynamics of CCSs, we generated kymographs of large CLC-positive structures detected in live-cell TIRF movies. In these kymographs, plaques showed quick and large fluctuations in shape and density (Fig. 5a). In some particular cases, clear disassembly into smaller structures could also be observed (Fig. 5a).

Strikingly, visual inspection of 3D SR images of control HeLa cells revealed CCPs and/or vesicles appearing on top of plaques (Figs 1b and 5b, Supplementary Movie 2). These observations are consistent with EM studies[19,25] that have captured pits seemingly emerging at the periphery of plaques. Thus, plaques may be intermediates in the formation of vesicles.

We reasoned that if CCVs emerge from plaques, inhibition of their scission would produce accumulation of pits on the membrane at the cost of plaques. Thus, we treated control HeLa cells with Dynasore, a small molecule inhibitor of dynamin that prevents vesicle scission and causes the accumulation of elongated pits on the plasma membrane[39]. Strikingly, 3D SR pictures of Dynasore-treated cells displayed a substantial reduction of plaques at the basal membrane and abundant pits, which often formed clusters (Fig. 5c). Morphometric analysis of 2D SR images of cells treated with DMSO (mock) or Dynasore

further confirmed the markedly reduced area covered by plaques and the increase in pit number (Fig. 5d,e).

As acute inhibition of dynamin results in the buildup of clumps of pits at the expense of plaques, plaques appear to be hubs where pits can form and pinch off. Furthermore, the accumulation of plaques upon depletion of either N-WASP or the Arp2/3 complex suggests a plaque-specific role for actin polymerization in the formation of these pits.

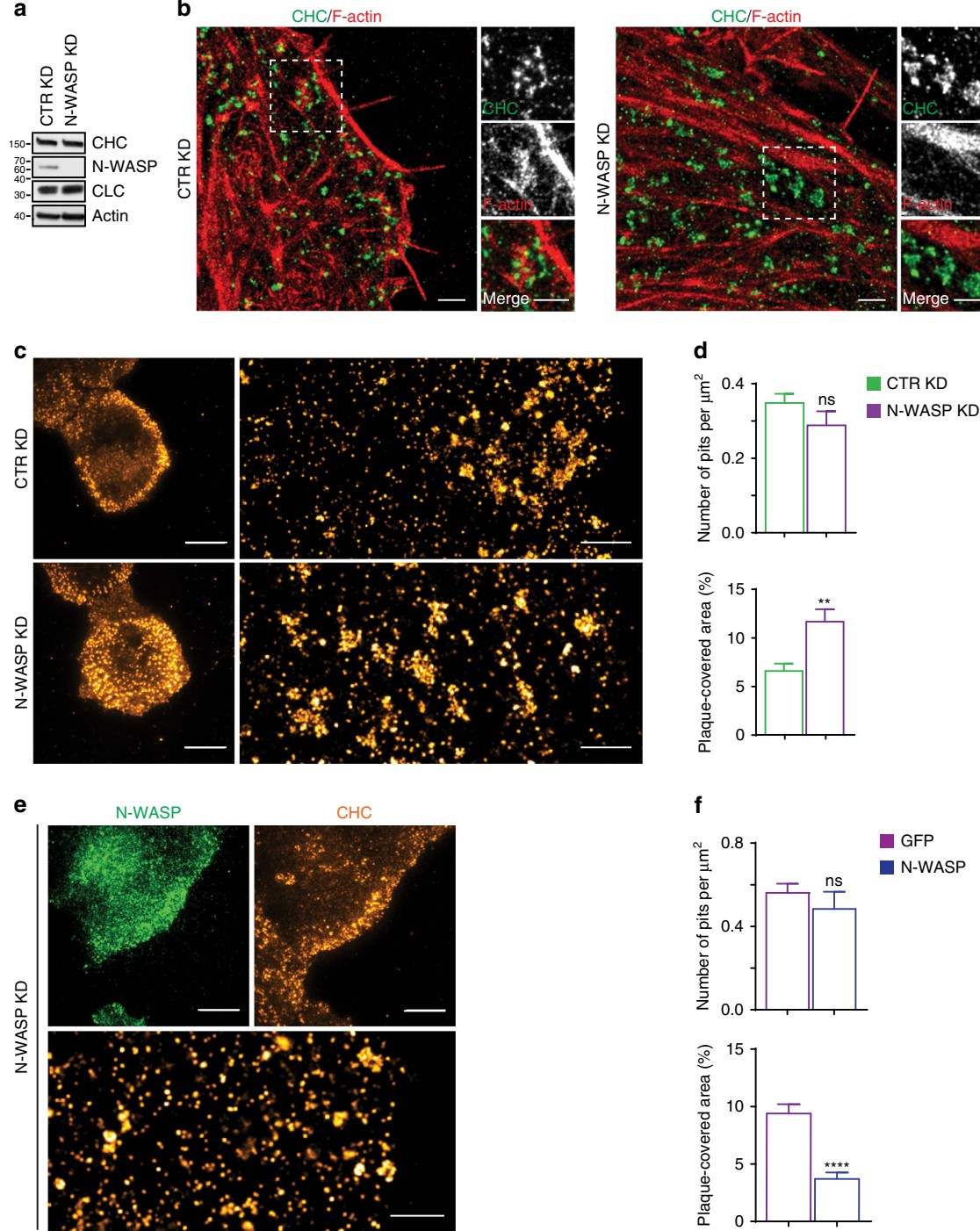

**Figure 2 | N-WASP is a key plaque regulator.** (**a**) Characterization of control (CTR) knockdown (KD) and N-WASP KD HeLa cells. Total cell lysates were compared using the indicated antibodies. One of three experiments that were performed with similar results is shown. (**b**) N-WASP regulates the association between CCSs and F-actin. Representative two-colour SR images of control (CTR) KD and N-WASP KD cells stained for CHC (green in merge) and F-actin (red in merge). Dashed white boxes mark the position of the zoom-in. Scale bar, 1 μm. (**c**) Knockdown of N-WASP increases plaque presence on the basal membrane. Representative TIRF and SR images of CHC on the basal membrane of control (CTR) KD and N-WASP KD cells. Scale bar TIRF, 10 μm. Scale bar SR, 1 μm. (**d**) Bar graphs show the number of pits per μm$^2$ and percentage of total area of the ROI covered by plaques (plaque covered area). ROIs were defined as three μm-wide regions on the periphery of the cells and segmented for quantification (mean ± s.e.m., **$P < 0.01$, $t$-test, ns = not significant). (**e**) Representative TIRF and SR images of CHC in N-WASP KD cells transfected with full-length shRNA-resistant N-WASP tagged with GFP (N-WASP-GFP). Scale bar TIRF, 10 μm; SR, 1 μm. (**f**) Number of pits and plaque-covered area were obtained as above (mean ± s.e.m., ****$P < 0.0001$, $t$-test, ns, not significant).

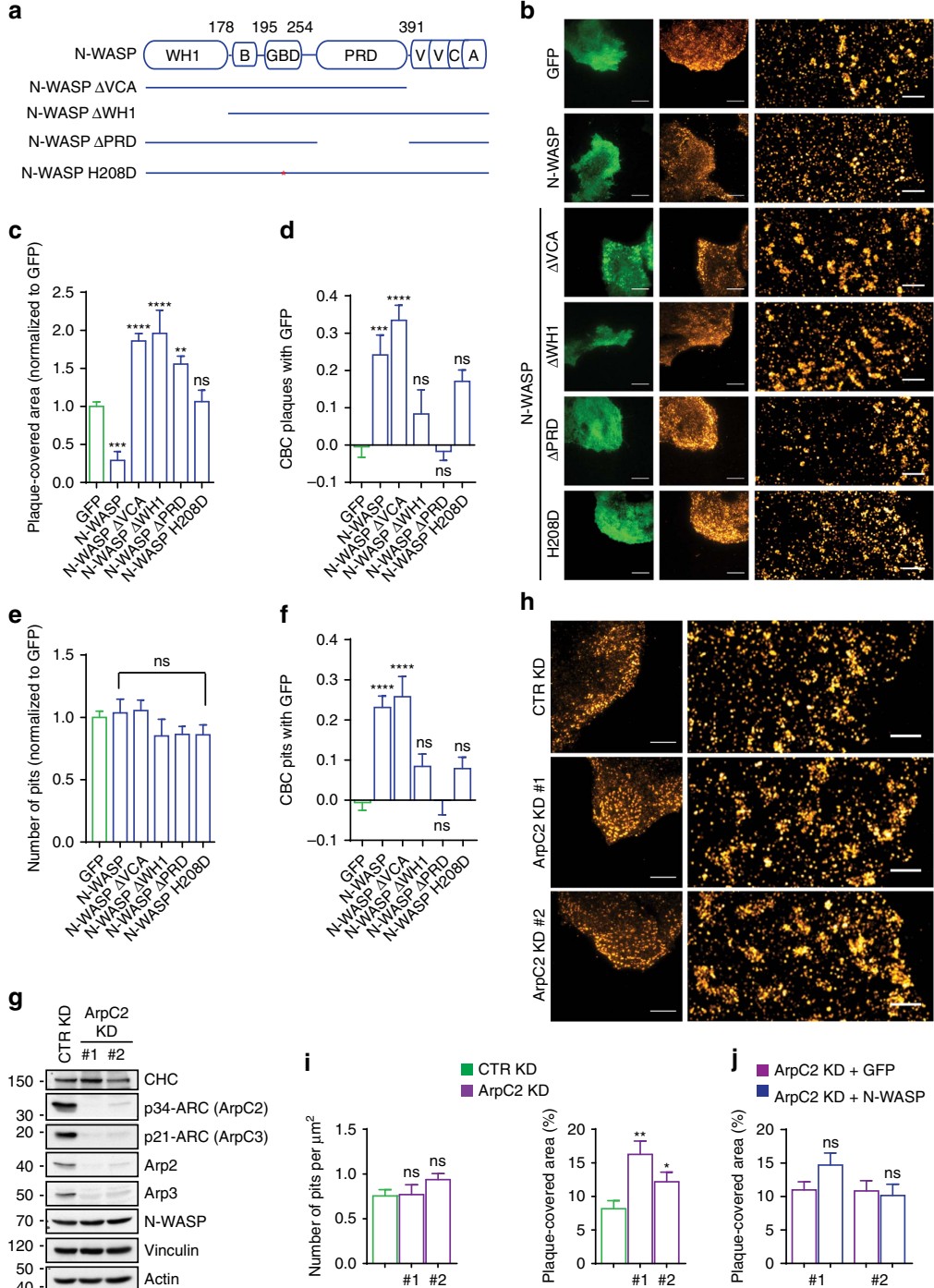

**Figure 3 | Actin polymerization controlled by N-WASP and the Arp2/3 complex has a key role in regulating plaques but not pits.** (**a**) Schematic of N-WASP and mutants thereof (asterisk = H208D mutation). Supplementary Fig. 1c decodes abbreviations. (**b**) Representative TIRF and SR images of CHC on the basal membrane of cells expressing the GFP-tagged N-WASP mutants described in **a**. Scale bar TIRF, 10 μm; SR, 1 μm. (**c**) Active N-WASP regulates plaques through its VCA, WH1 and PRD domains. Bar graph shows normalized plaque covered area (mean ± s.e.m., \*\*$P < 0.01$, \*\*\*$P < 0.001$, \*\*\*\*$P < 0.0001$, one-way ANOVA, ns, not significant). (**d**) Localization of N-WASP at plaques requires active N-WASP, the WH1 and the PRD domains. Bar graph shows coordinate-based colocalization (CBC) for the association between plaques and the N-WASP mutants or GFP (mean ± s.e.m., \*\*\*$P < 0.001$, \*\*\*\*$P < 0.0001$, one-way ANOVA). (**e**) Neither N-WASP nor its mutants perturb pits. Bar graph shows normalized number of pits (mean ± s.e.m., one-way ANOVA, ns = not significant). (**f**) Localization of N-WASP at pits requires active N-WASP, the WH1 and the PRD domains. Bar graph shows the CBC for the association between pits and the N-WASP mutants or GFP (mean ± s.e.m., \*\*\*\*$P < 0.0001$, one-way ANOVA, ns = not significant). (**g**) Characterization of ArpC2 KD HeLa cells. Control (CTR) KD and ArpC2 KD cells (#1 and #2 obtained using different hairpins) were compared using the indicated antibodies. One of two similar experiments is shown. (**h**) The Arp2/3 complex controls CCS morphology. Representative TIRF and SR images of CHC on the basal membrane of the above-characterized cells. Scale bar TIRF, 10 μm; SR, 1 μm. (**i**) Knockdown of the Arp2/3 complex phenocopies that of N-WASP. Bar graphs show number of pits per μm$^2$ and plaque covered area (mean ± s.e.m., \*$P < 0.05$, \*\*$P < 0.01$, one-way ANOVA, ns = not significant). (**j**) N-WASP regulates plaques through the Arp2/3 complex. Bar graph shows plaque covered area in ArpC2 KD cells transfected with N-WASP-GFP or GFP (mean ± s.e.m., one-way ANOVA, ns = not significant).

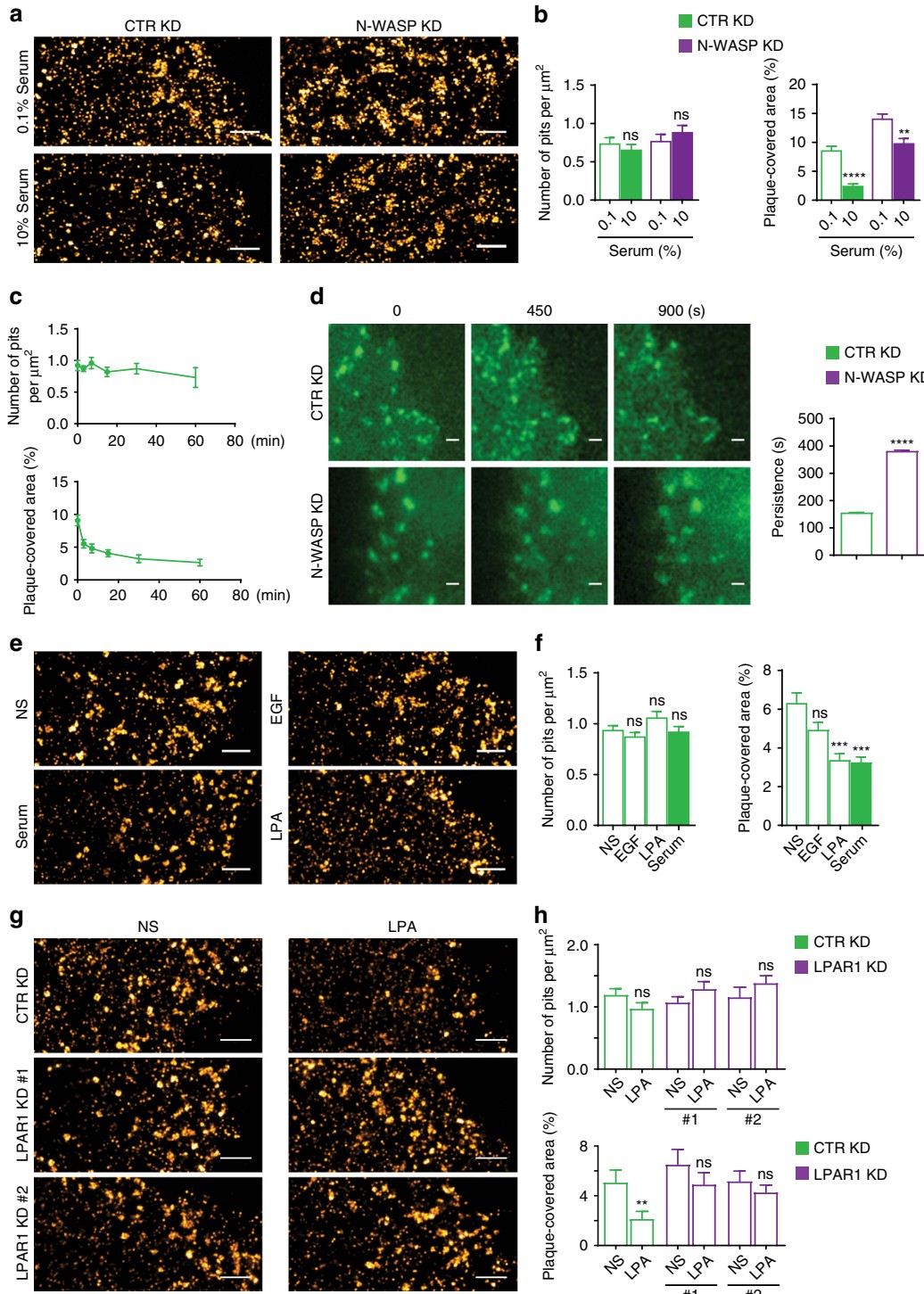

**Figure 4 | Plaques remodel in response to external stimuli.** (**a**) Representative SR images of the basal membrane of control (CTR) KD and N-WASP KD HeLa cells grown in either 0.1 or 10% serum and stained with anti-CHC antibodies to detect CHC. Scale bar, 1 μm. (**b**) Serum affects the presence of plaques, but not that of pits. Bar graph shows number of pits per μm$^2$ and plaque covered area (percentage) of cells grown in either 0.1 or 10% serum (mean ± s.e.m., **$P < 0.01$, ****$P < 0.0001$, $t$-test, ns = not significant). (**c**) Plaques rapidly respond to serum stimulation. Bar graph shows number of pits per μm$^2$ and plaque covered area of serum-starved cells stimulated with 10% serum for 3, 7, 15, 30 and 60 min (mean ± s.e.m.). (**d**) The knockdown of N-WASP increases the persistence of CCSs in cells stimulated with serum. Representative images of live-cell TIRF movies of control (CTR) KD and N-WASP KD cells expressing CLC-RFP, stimulated with 10% serum at time 0. Bar graph shows CCS track duration (sec. = seconds, mean ± s.e.m., ****$P < 0.0001$, $t$-test). Scale bar, 1 μm. (**e**) LPA recapitulates the effects of serum on plaques. Representative SR images control (CTR) KD cells that were serum starved and stimulated with 100 ng ml$^{-1}$ EGF, 5 μM LPA or 10% FCS for 30 min or left untreated (NS) and subsequently stained for CHC. Scale bar, 1 μm. (**f**) Bar graphs show number of pits per μm$^2$ and plaque covered area (mean ± s.e.m., ***$P < 0.001$, one-way ANOVA, ns = not significant). (**g**) LPAR1 mediates the effects of LPA on plaques. Representative SR images of CHC on the basal membrane of control (CTR) KD and LPAR1 KD HeLa cells (#1 and #2 obtained using different hairpins). Scale bar, 1 μm. (**h**) Bar graphs show number of pits per μm$^2$ and plaque covered area (mean ± s.e.m., **$P < 0.01$, $t$-test, ns = not significant).

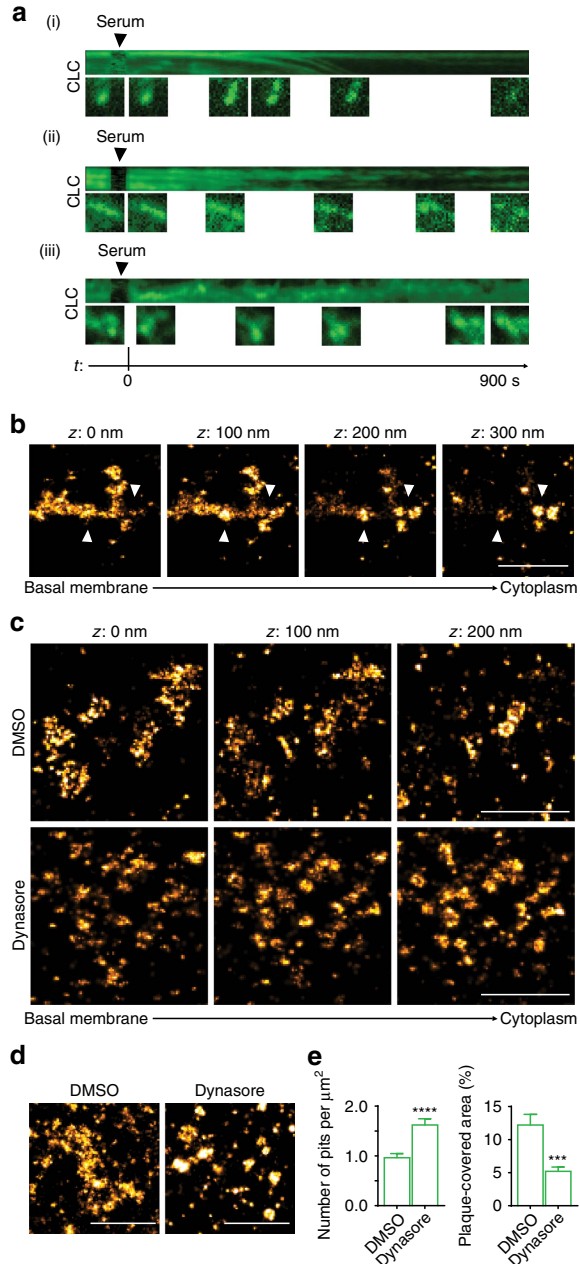

**Figure 5 | Plaques are sites of pit formation.** (**a**) CCSs exhibit different dynamics and fate. Three representative kymographs of large CCSs from live-cell TIRF movies of HeLa cells expressing CLC-RFP that were serum starved overnight and then stimulated with 10% serum. CCSs can dissociate into smaller structures (i and ii) or fluctuate (iii). Time (t) is in seconds (s). Representative stages are shown below each kymograph. (**b**) Plaques are sites of pit formation. Two representative 3D SR image slices of HeLa cells stained for CHC showing CCVs (whose position is marked by arrowheads in the Z stacks) in areas covered by plaques. Scale bar, 1 μm. (**c**) Inhibition of dynamin perturbs the morphology of CCSs. Representative 3D SR image slices of HeLa cells treated with DMSO or the dynamin inhibitor Dynasore (80 μM, 30 min) stained for CHC. Arrowheads mark pits in the Z stacks. Scale bar, 1 μm. (**d**) Representative SR images of HeLa cells treated with DMSO or the dynamin inhibitor Dynasore (80 μM, 30 min) stained for CHC. Scale bar, 1 μm. (**e**) Inhibition of dynamin reduces plaques and increases pit number. Bar graph shows number of pits per μm² and plaque covered area (percentage) (mean ± s.e.m., ***P < 0.001, ****P < 0.0001, t-test).

**Plaques are hubs for CME and signalling of LPAR1.** Given that plaques are sites of pit assembly and disappear upon LPA stimulation, they could participate in ligand-induced CME. Hence, we determined the localization of endogenous EGFR and GFP-tagged LPA receptor 1 (LPAR1-GFP) in control and N-WASP KD cells counterstained for CHC using two-colour SR. LPAR1 and EGFR attained a diffuse and homogeneous distribution on the basal plasma membrane of both cell lines before activation by the cognate ligand (Fig. 6a,b). Activation of LPAR1 with LPA and EGFR with EGF triggered receptor recruitment to plaques and pits (Fig. 6a,b), supporting that internalization of these receptors occurs primarily via CME[4,16]. Consistently, ligand-induced internalization of EGFR and LPAR1 were impaired by downregulating N-WASP (Fig. 6c). However, only LPA caused the disappearance of plaques from the basal membrane of control cells (Fig. 4e), which suggests that the recruitment of the LPAR1 to plaques is followed by internalization.

If plaques were sites of CME of LPAR1, one may expect that (i) the dissolution of plaques induced by LPA would require LPAR1 and (ii) reducing LPAR1 internalization in N-WASP KD cells could perturb, at least partly, the activity of signal transduction pathways downstream of LPAR1. We silenced the LPAR1 in HeLa cells to test the first prediction (Supplementary Fig. 5) and found that this significantly attenuated LPA-induced plaque disassembly with respect to the control KD cells (Fig. 4g,h). To test the second prediction, we compared the activity of ERK and AKT pathways, the two main signalling axes downstream of LPAR1 and EGFR, in control KD and N-WASP KD cells. The activity of ERK and AKT induced by EGF did not depend on N-WASP, the area covered by plaques or the dynamics thereof (Fig. 7a). In contrast, LPA signalling to AKT was dramatically increased upon knockdown of N-WASP (Fig. 7b). This suggests that in the absence of proper internalization, LPA receptors transmit stronger and/or more prolonged signals. To study this at higher temporal resolution, we made use of a specific biosensor to monitor the production of phosphatidylinositol 3,4,5 trisphosphate (PIP$_3$), an upstream activator of AKT. Membrane relocalization of the pleckstrin homology (PH) domain of GRP1 (ref. 40) induced by LPA stimulation was significantly higher in N-WASP KD cells than in the control cells (Fig. 7c). Instead, no significant difference was detected when EGF was used (Fig. 7c). Hence, increased LPA-induced PIP$_3$ formation in N-WASP KD cells translates into higher levels of activated AKT (Fig. 7b). The effects of LPA are mediated by LPAR1 because the small molecule inhibitor Ki6425, which blocks both LPAR1 and LPAR3 (ref. 41), abrogated AKT activation without perturbing CCSs in resting cells (Supplementary Fig. 6a,b) and LPAR3 is not expressed in HeLa cells[42]. These results suggest that LPAR1, but not EGFR, is internalized from plaques and raise the intriguing possibility that plaques may function as signaling nanodomains.

An independent line of evidence supports this view: live-cell TIRF experiments showed that the LPAR1 was rapidly recruited to CCSs after LPA stimulation in both control KD and N-WASP KD cells (Fig. 7d,e). In the control KD cells, the marked colocalization of LPAR1 and CCSs decreased progressively and was accompanied by LPAR1 endocytosis, as indicated by the accumulation of LPAR1-GFP-positive vesicles inside the cells (Fig. 7d,e and Supplementary Movie 3). In the N-WASP KD cells, LPAR1-GFP persisted for several minutes in CCSs and it was not prominently internalized after LPA stimulation (Fig. 7d,e and Supplementary Movie 4). Moreover, two-colour kymographs of large CCSs in the control KD cells revealed co-fluctuation of CLC-RFP and LPAR1-GFP (Fig. 7f). In contrast, in kymographs from N-WASP KD cells CLC and LPAR1 colocalized more persistently (Fig. 7f). Notably, no evident difference was observed

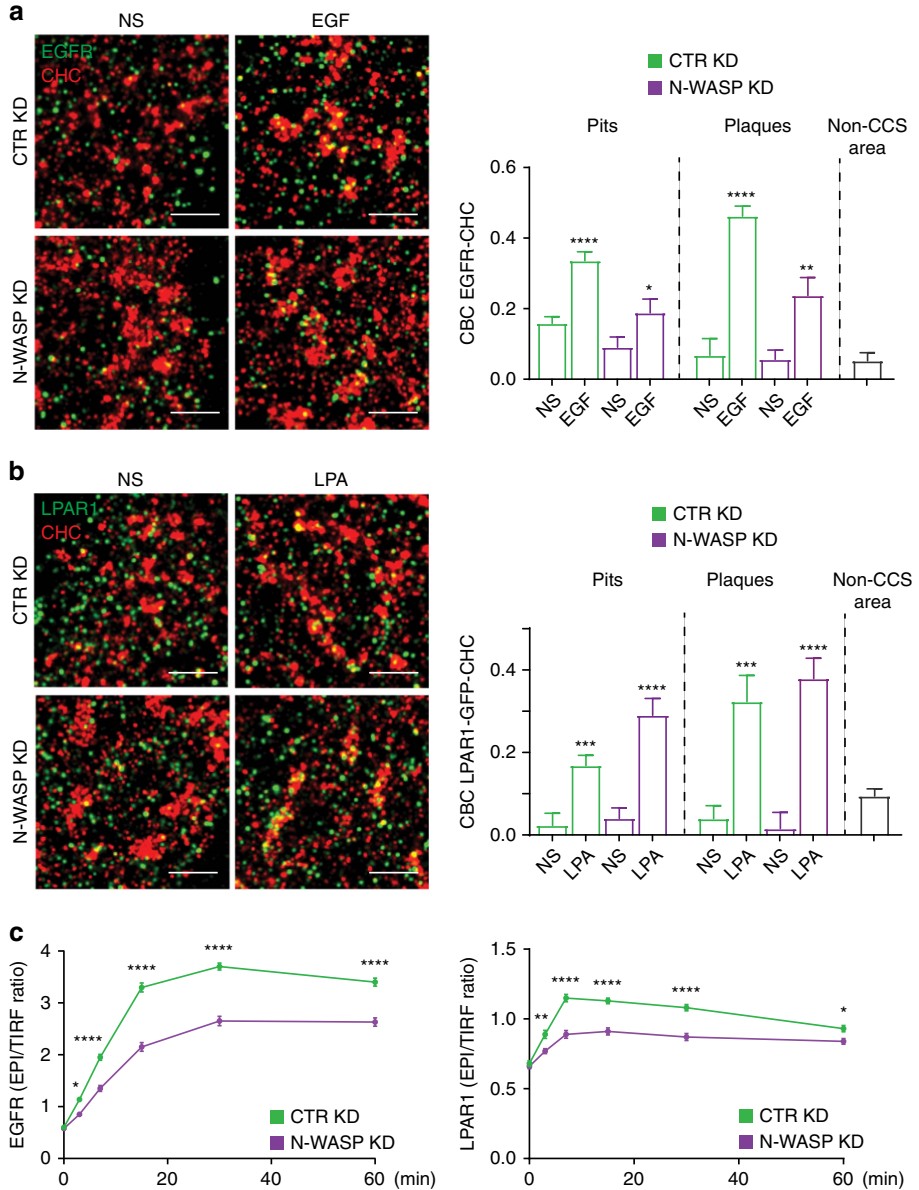

**Figure 6 | LPAR1 and EGFR are recruited to plaques. (a)** Activated EGFR is recruited to CCSs. Representative two-colour SR images of control (CTR) KD and N-WASP KD cells stained for CHC (red in merge) and EGFR (green in merge). Cells were stimulated with 100 ng ml$^{-1}$ EGF for 5 min. Scale bar, 1 µm. Bar graph shows CBC for the association of EGFR with either pits or plaques before and after EGF stimulation (mean ± s.e.m., *$P<0.05$, **$P<0.01$, ****$P<0.0001$, $t$-test). **(b)** Activated LPAR1 is recruited to CCSs. Representative two-colour SR images of control (CTR) KD and N-WASP KD cells stably expressing LPAR1-GFP stained for CHC (red in merge) and GFP (green in merge). Cells were stimulated with 5 µM LPA for 5 min. Scale bar, 1 µm. Bar graph shows CBC for the association of EGFR with either pits or plaques before and after LPA stimulation (mean ± s.e.m., ***$P<0.001$, ****$P<0.0001$, $t$-test). **(c)** Internalization of EGFR and LPAR1 is impaired in cells lacking N-WASP. Graphs show internalization level of EGFR and LPAR1 measured as the ratio of EPI images versus TIRF images of control (CTR) KD and N-WASP KD cells after stimulation with EGF or LPA, respectively, for 3, 7, 15, 30 and 60 min (mean ± s.e.m., *$P<0.05$, **$P<0.01$, ***$P<0.001$, ****$P<0.0001$, one-way ANOVA).

in the behaviour of pits in the two cell lines (Supplementary Movies 3 and 4). Therefore, increased PIP$_3$ formation and AKT activation after LPA stimulation in cells with impaired plaque dynamics result from the lingering of the receptor in these structures. Consistent with this notion, LPA activated AKT more prominently in the Arpc2 KD cells than in control cells (Fig. 7g). Global actin depolymerization by Latrunculin A recapitulated the increase in plaques observed upon knockdown of either N-WASP or the Arp2/3 complex, but it also severely depleted the cellular pit pool (Supplementary Fig. 7a). Surprisingly, Latrunculin A dampened both LPA-induced and EGF-induced activation of

AKT, regardless of N-WASP expression (Supplementary Fig. 7b). Hence, optimal AKT activation seemingly involves an additional N-WASP-independent, yet actin-sensitive event that occurs upstream of or parallel to plaque regulation.

## Discussion
Here we took advantage of the increased resolution provided by SR and combined it with molecular genetics and cell biology to unravel fundamental functional properties of flat clathrin-coated plaques that distinguish them from pits.

We find that plaques function as dynamic hubs for actin-dependent assembly of CCVs that regulate signalling and endocytosis of LPAR1, but not EGFR. At variance with the pits that form *de novo* on the plasma membrane, pit maturation at plaques requires N-WASP and its ability to promote Arp2/3-complex-dependent actin polymerization. Consistently, we identified unique requirements for the action of N-WASP at

plaques. First, the ΔPRD mutant markedly increased the plaque-covered area, whereas it did not affect the pit number (Fig. 3) or cell-surface EGFR[7]. Second, downregulation of the N-WASP activator Abi1 in Nap1 KD cells, which impairs CME of the EGFR[7], had no effect on plaques (Supplementary Fig. 8). This indicates the existence of a plaque-specific set of N-WASP regulators. Instead, a common mechanism relying on the

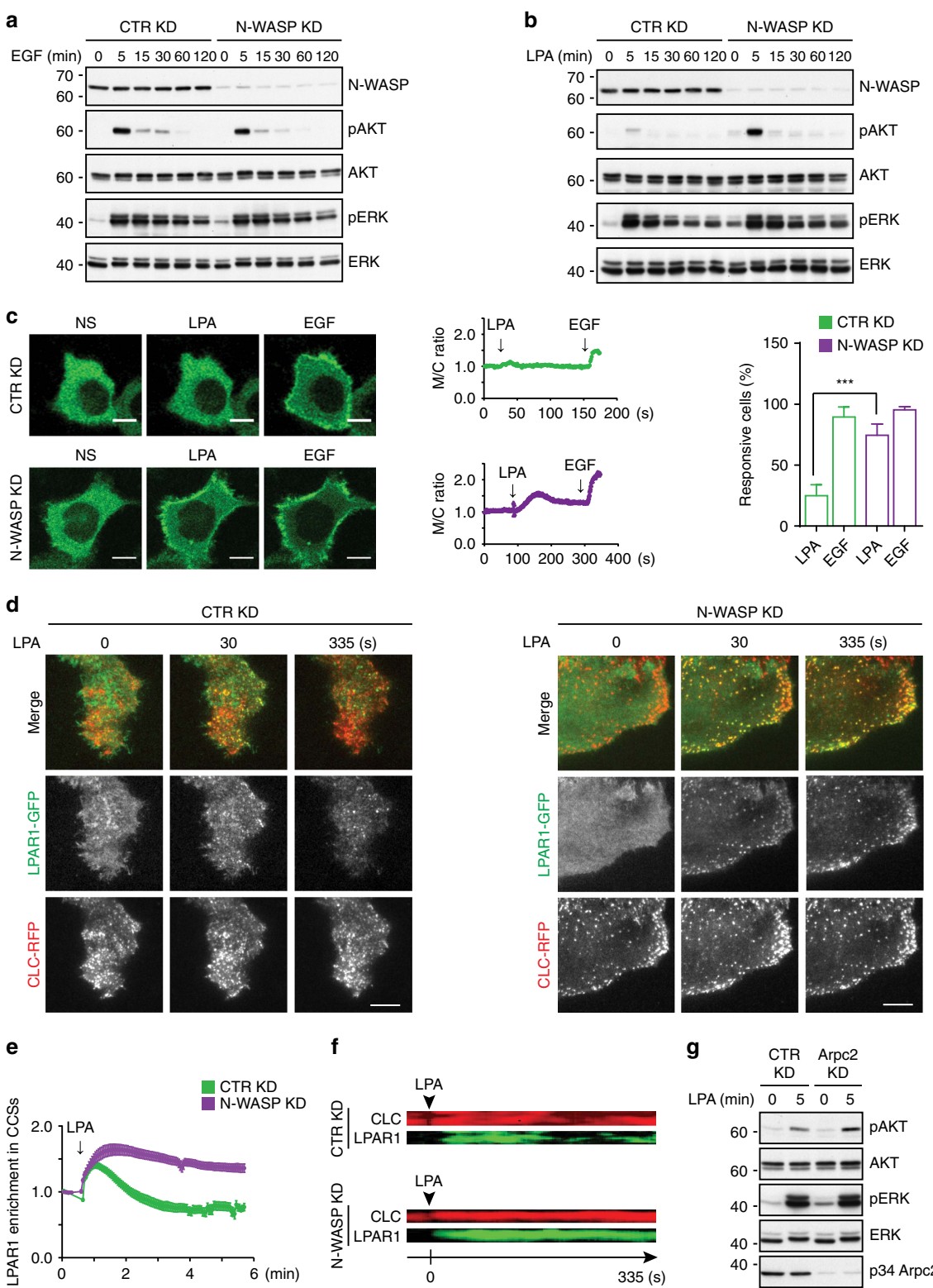

WH1 domain, the GBD and the central PRD controls the recruitment of N-WASP to pits and plaques, which both contain the CME machinery. Interestingly, these domains have a well-established role in regulating N-WASP auto-inhibition. Thus, the enrichment of N-WASP ΔVCA in both CCS types, vis-à-vis the absence of the H208D mutant, jointly suggest that N-WASP stably associates with pits and plaques only in its open conformation.

Our finding that the LPAR1 is internalized upon recruitment to plaques favours the idea that CCVs can form through a process that involves remodelling of flat plaques. Although a different and widely accepted model for CCV formation holds that pits are initiated *de novo* at endocytic sites and maintain a constant curvature throughout their maturation[43], four independent lines of evidence support our view. First, high-quality EM micrographs depict pits and vesicles surrounding or even emerging from plaques[5,18,19,25]. Second, computer simulations suggest that conversion of clathrin triskelia from a flat to a curved basket-like arrangement is possible both from a thermodynamic and a molecular point of view[44]. Third, correlative light electron microscopy shows that the clathrin coat of pits may start from a rather flat configuration[45]. Fourth, non-terminal scission events at long-lived CCSs have been observed in live cells[46].

The essential role of actin polymerization in CCV formation at plaques discovered herein likely consists in providing mechanical force for the rearrangement of the flat clathrin lattice into a curved lattice and/or in counteracting the resulting increase in membrane tension. Consistent with this notion, drugs that arrest actin dynamics inhibit the formation of CCVs and prolong CCS lifetime[5,6]. In correlative light electron microscopy studies, thin actin fibres visible in EM are not always detected in the corresponding fluorescence images[12]. Thus, the presence of F-actin at plaques might be a general feature of these clathrin nanodomains that has gone unnoticed using low-resolution microscopy approaches.

In light of the above, it is not surprising that cell-substrate adhesion and membrane tension affect CCS formation and dynamics[21,47]. As strong adhesion delays pit dynamics and CCV formation[21,47] and increases the abundance of plaques on the ventral, but not the dorsal side of cells[21,43,47], it has been claimed that plaques may be tissue-culture artefacts. However, plaques have been observed in physiological contexts greatly varying for adhesion and tension, such as osteoclasts and bone, myocytes, non-adhering adipocytes and even on the non-adherent surface of cultured cells[22,35,48,49]. We observed that plaques do not dissolve when actomyosin-mediated cellular tension is reduced by treatment with Latrunculin A or ROCK inhibitor Y-27632 (Supplementary Fig. 7a–c). Moreover, plaques showed little overlap with the abundant basal stress fibres formed by the N-WASP KD cells (Fig. 2b and Supplementary Fig. 3c). Therefore, it seems unlikely that actin-based adhesion strengthening is a major determinant of plaques.

The ability of plaques to serve as dynamic actin-controlled hubs for CME of the LPAR1 has profound effects on cell signalling. Interestingly, the prolonged residence of LPAR1 at plaques in the N-WASP KD cells correlates with higher amplitude of AKT, but not ERK, activity. This likely reflects the fact that AKT activation depends on $PIP_3$, which is generated by activated PI3K using $PIP_2$ as a substrate, and the availability of $PIP_2$ at high concentration at sites of pit formation[50].

Yet, we discover that plaques *per se* seem to inhibit cell migration. We found that the N-WASP KD cells moved less than the control ones in the absence of external cues and after addition of EGF (Supplementary Fig. 9a). Of note and at variance with LPA, EGF does not cause plaque dissolution (Fig. 4e) and induces AKT and ERK activation and membrane ruffling independently of N-WASP (Fig. 7)[7]. Thus, plaque abundance scales inversely with cell movement independently of the activation of extracellular signal-regulated pathways. As plaque abundance had no effect on either integrin-dependent or integrin-independent cell adhesion (Supplementary Fig. 9b), the exact mechanism of this anti-migratory role remains to be investigated.

Finally, the importance of LPA for cancer cell chemotaxis[51,52] and the correlation between high expression of N-WASP and cancer dissemination[53] suggest that LPA-dependent control of plaques may be an unforeseen mechanism whereby N-WASP exerts its pro-metastatic function.

## Methods

**Chemicals and reagents.** High-glucose DMEM supplemented with pyruvate and GlutaMax was from Invitrogen. Protease inhibitor EDTA-free cocktail, X-tremeGene 9 were from Roche. Gelatine with ∼300 bloom, Glucose Oxidase Type VII from *Aspergillus niger*, Catalase from *Aspergillus niger*, EGTA, MES, Dynasore, DMSO, Y27652, Sodium borohydride, Latrunculin A and all other chemicals if not otherwise specified were from Sigma-Aldrich. Fetal bovine serum was from APS. LPA (1-oleyl) was from Avanti Polar Lipids, EGF was from Invitrogen, Ki16425 was from Santa Cruz biotechnology. Cysteamine hydrochloride-MEA was from Fluka. PIPES was from Fisher scientific. Paraformaldehyde, glutaraldehyde solution 25%, Magnesium chloride hexahydrate, Triton X-100 were from Merck. Albumin bovine Fraction V, pH 7.0 was from Serva.

**Antibodies.** Antibodies were as follows: mouse anti-β-actin (AC-15) (A5441, Sigma-Aldrich, WB 1:2,000), mouse anti-CLC (CON.1) (ab24579, Sigma-Aldrich, WB 1:1,000), mouse anti-CHC (X-22; ABR) (MA1-065, Invitrogen, IF 1:200), goat anti-ARP2 (N-15) (sc-10125, Santa Cruz Biotechnology, WB 1:500), goat anti-ARP3 (G-15) (sc-10130, Santa Cruz Biotechnology, WB 1:500), goat anti-p34-ARPC2 (IMG-3497, Imgenex, WB 1:500), mouse anti-p21-ARPC3 (ab96137, Transduction Laboratories, WB 1:500), rabbit anti-RFP polyclonal (AB3216, Chemicon, IF 1:200), mouse anti-vinculin (ab18058, Abcam, WB 1:1,000), rabbit anti-GFP polyclonal was a kind gift from G. Kops, mouse anti-EGFR (Ab-1 528, Calbiochem, IF 1:50), rabbit

**Figure 7 | Plaques are hubs for clathrin-mediated endocytosis and signalling of LPAR1.** (**a**) Knockdown of N-WASP does not perturb the activation of AKT or ERK induced by EGF. Serum-starved control (CTR) KD and N-WASP KD HeLa cells were left untreated (0) or stimulated with 100 ng ml⁻¹ EGF as indicated. Total cell lysates were analysed with the indicated antibodies. One of three similar experiments is shown. (**b**) Knockdown of N-WASP induces hyper-activation of AKT after LPA stimulation. Serum-starved control (CTR) KD and N-WASP KD HeLa cells were left untreated (0) or stimulated with 5 μM LPA as indicated. Total cell lysates were analysed with the indicated antibodies. One of three similar experiments is shown. (**c**) Knockdown of N-WASP increases $PIP_3$ formation after LPA but not EGF stimulation. Representative $PIP_3$ formation tracking images using a $PIP_3$ sensor (GRP1) tagged with GFP in live cell confocal images of control (CTR) KD and N-WASP KD HeLa cells stimulated with 100 ng ml⁻¹ EGF and 5 μM LPA. Scale bar, 10 μm. Representative traces (one cell per trace) of brightness ratio between membrane and cytoplasm (M/C ratio). Bar graph shows percentage of responsive cells (mean ± s.e.m., \*\*\*P<0.001, t-test). (**d**) Activated LPAR1 is recruited to CCSs, and its internalization is impaired in cells lacking N-WASP. Representative images of two-colour live cell TIRF movies of control (CTR) KD and N-WASP KD cells expressing CLC-RFP (red in merge) and LPAR1-GFP (green in merge), before and after stimulation with 5 μM LPA. Scale bar, 10 μm. (**e**) Graph shows LPAR1-GFP enrichment in CLC-RFP positive structures in control (CTR) KD and N-WASP KD cells. (**f**) Representative kymographs of non-diffraction limited structures selected in the CLC-RFP channel of two-colour live cell TIRF movies showing CLC-RFP (red) and LPAR1-GFP (green) of control (CTR) KD and N-WASP KD HeLa cells, before and after stimulation with 5 μM LPA. (**g**) Knockdown of Arpc2 induces hyper-activation of AKT after LPA stimulation. Serum-starved control (CTR) KD and Arpc2 KD cells were stimulated with 5 μM LPA for 5 min. Total cell lysates were analysed with the indicated antibodies. One of two similar experiments is shown.

anti-AKT (C67E7, Cell Signaling, WB 1:1,000), rabbit anti-pAKT (Ser473) (D9E, Cell Signaling, WB 1:1,000), rabbit anti-p44/42 ERK (20G11, Cell Signaling, WB 1:1,000) and anti-ERK (9102, Cell Signaling, WB 1:1,000), rabbit anti-Integrin β5 (3629, Cell Signaling, IF 1:1,000) and rabbit anti-N-WASP (30D10, Cell Signaling, WB 1:1,000). Alexa-fluor-532 and Alexa-fluor-647 conjugated goat anti-mouse and anti-rabbit IgG (H + L) antibodies (A-11002, A-11009, A-32728, A-32733, Invitrogen, 1:200 dilution). Alexa-fluor-647 phalloidin (A22287, Invitrogen, 1:50 dilution).

**Expression vectors.** Rat N-WASP (1-501) constructs were either previously described[7,8] or generated by PCR amplification and sequence verified. Primer sequences are listed in Supplementary Table 1. mRFP1-CLC was from L. Lagnado, mTurquoise2-CLC was from D. Gadella. LPAR1-GFP was from W. Moolenaar. pGRP1(PH)-EGFP was from A. Gray[40].

**Cell culture and procedures for transfection and knockdown.** HeLa cells were cultured in DMEM GlutaMax supplemented with 10% FCS. BSC-1 cells were cultured in MEM (PAA) supplemented with 10% FCS and GlutaMax. Cells were transfected with X-tremeGene9 according to the manufacturer's instructions. Stable N-WASP knockdown HeLa cells were previously described[7]. Stable N-WASP BSC-1 cells were obtained by lentiviral infection with MISSION TRC shRNA TRCN0000123061 (#1) and TRCN0000123063 (#2). Arpc2 knockdown HeLa cells were obtained by lentiviral infection with MISSION TRC shRNA TRCN0000036502 (#1) and TRCN0000036503 (#2) using previously described procedures[54,55]. Stable Nap1 KD HeLa cells have been previously generated and characterized[37]. Stable LPAR1-GFP HeLa cells were obtained by retroviral infection and selected based on GFP expression levels by FACS sorting. Stable LPAR1 KD HeLa cells were obtained by retroviral infection with MISSION TRC shRNA TRCN0000036502 (#1) and TRCN0000036503 (#2) using previously described procedures[54,55].

**RT-qPCR.** Total RNA were isolated, retrotranscribed and used for qPCR analyses using standard procedures as previously described[56,57]. Primers for LPAR1 and GADPH have been previously described and validated[52,56,57].

**Western blotting.** Cells were cultured in six-well plates, serum starved overnight in medium containing 0.1% FCS, treated with inhibitors (Ki16425 1 μM for 10 min) and stimulated with agonists as indicated. Whole-cell lysates were prepared by scraping PBS-washed cells in JS lysis buffer[37,57] supplemented with $Na_3VO_4$ 5 μM, NaF 1 μM and protease inhibitor cocktail (Roche). Membranes were blocked in BSA 5% in TBS supplemented with Tween-X20 0.2% and incubated with primary antibodies, followed by HRP-conjugated secondary antibodies. Full scans of western blots can be found in Supplementary Fig. 11.

**Live-cell confocal and TIRF imaging.** HeLa and BSC-1 cells were seeded on 24 mm coverslips (#1.5) coated with 0.5% gelatine/PBS unless otherwise specified. Cells were transfected for 2–3 h and serum starved overnight in medium containing 0.1% FCS. Cells were imaged in DMEM-F12 medium at 37 °C in a humidified chamber with 5% $CO_2$ level using a TCS SP8 confocal or an AM TIRF MC from Leica Microsystems.

**Electron microscopy.** Cell unroofing and shearing was performed as previously described[58] to expose the cytoplasmic face of the ventral plasma membrane of cells adhering to gelatine-coated coverslips. Briefly, samples were rinsed briefly with ice-cold 10 mM PIPES (pH 6.0) and then incubated for 3–5 min at room temperature in the same buffer supplemented with 3 mM $ZnCl_2$, 60 mM NaCl and 0.67 mM KCl in order to open the cells. Next, cell shearing was obtained by spraying ice-cold 50 mM PIPES buffer (pH 6.0) supplemented with 10 mM $MgCl_2$ onto the entire surface of coverslips using a syringe. After the cells had been sheared open mechanically, the ventral membrane and its associated cytoskeletal elements were fixed in 2% glutaraldehyde for 30 min followed by dehydration in ethanol series. The final step consisted in changing 100% ethanol by Hexamethyldisilazane and letting this dry in the air. Dried cells were gold sputtered and viewed in a Zeiss SEM.

**Adhesion and migration assays.** For cell adhesion assays, cells were trypsinized and washed three times with DMEM supplemented with 0.1% BSA. Cells ($10^5$) were seeded into 96-well plates coated with 2 μg ml$^{-1}$ Collagen I for 30 min at 37 °C, or poly-lysine (0.1 mg ml$^{-1}$) for 5 min at 37 °C, or mock coated with PBS, and blocked with blocking buffer (DMEM supplemented with 0.5% BSA). After incubation for 1 h at 37 °C, non-adherent cells were removed by five washes using washing buffer (DMEM supplemented with 0.1% BSA). Attached cells were fixed with PFA for 10 min, washed twice and stained with Crystal Violet (5 mg ml$^{-1}$ in 2% ethanol) for 10 min. Fixed cells were extensively washed with water and lysed with 2% SDS for 30 min. Plates were read at 490 nm (ref. 59). Absorbance values obtained from mock-coated wells were considered as background and subtracted from all other raw values.

For migration assays, cells were plated on 0.5% gelatine/PBS-coated six-well multiwell plates and, after 24 h, serum starved overnight in medium containing 0.1% FCS. Addition of SiR-DNA (Spirochrome) was performed 2 h prior to that of the indicated agonist and enabled nuclear tracking. Agonist-stimulated or non-stimulated cells were imaged every 5 min for 8 h using Zeiss Axio Observer Z1 microscope (Carl Zeiss) equipped with an LD Plan-Neofluar × 10 objective humidified climate chamber with 5% $CO_2$ at 37 °C, operated with Zeiss Microscope Software ZEN 2012. Movies were registered using the ImageJ plugin StackReg (http://imagej.net/StackReg) and, subsequently, nuclei were tracked using the ImageJ plugin TrackMate (http://imagej.net/TrackMate). Tracking was done using the LAP tracker algorithm, with a maximum travel distance of 15 μm per frame. Only tracks spanning the entire movie were quantified thereby excluding mitotic cells.

**GSDIM super-resolution imaging.** HeLa and BSC-1 cells were seeded on 24 mm coverslips (#1.5) coated with 0.5% gelatine/PBS unless otherwise specified. Cells were serum starved overnight in medium containing 0.1% FCS. Cells were washed briefly with DPBS and fixed as described before[31,55]. Briefly, 4% PFA in PEM buffer was added for 10 min, cells were then extracted in 0.5% Triton/PBS for 10 min and blocked for 1 h with 5% BSA/PBS. Inmunofluorescence was performed at room temperature with primary antibodies followed by secondary Alexa-fluor coupled antibodies. For actin colocalization studies Alexa-fluor-647 phalloidin was used. For Supplementary Fig. 10, cells were fixed for 10 min with methanol in PEM buffer pre-chilled at − 20 °C and blocked in BSA/PBS before staining, or with glutaraldehyde as described before[31,55]. Briefly, cells were fixed in 0.3% glutaraldehyde + 0.25% Triton X-100 for 2 min and in 0.5% glutaraldehyde for 8 min, diluted in cytoskeleton protective buffer (10 mM MES pH 6.1, 150 mM NaCl, 5 mM EGTA, 5 mM glucose, 5 mM $MgCl_2$. Cells were then incubated for 7 min in 0.1% $NaBH_4$ in PBS.

Imaging was carried out on a Leica SR-GSD 3D microscope. Images were taken in TIRF mode at 100 frames per second and 8,000–15,000 frames were collected. For 3D images, a cylinder lens[60] was used in conjunction with EPI mode imaging. Images were taken sequentially in decreasing excitation/emission wavelength order in the presence of an oxygen scavenging system (10% Glucose, 0.5 mg ml$^{-1}$ glucose oxidase, 40 μg ml$^{-1}$ Catalase) and 100 mM MEA). Data obtained this way were background subtracted[61] and localization of events, filtering and drift correction were made with the Thunderstorm[62] plugin for ImageJ. Images were rendered with 20-nm pixel size, with the Normalized Gaussian visualization option. For two-colour display, chromatic aberrations were corrected as detailed under 'Image Analysis'. For figure visualization, images were convolved with a mean filter of 3 × 3 pixels. For Supplementary Fig. 10c, images were analysed also using the default parameters of the analysis software provided by Leica LAS-AF and the open source software quickPALM[63] and rapidSTORM[64].

**Image analysis.** SR image quantification: plaques and pits were segmented and analysed with the particle analyser routine from Fiji (ImageJ)[65,66] with manually adjusted threshold. A 3 μm-wide region of interest (ROI) localized at the periphery of the cell was selected for analysis. The segmentation was done manually, since automated segmentation proved to be error-prone. Clathrin coated pits were classified as circular or donut-shaped clusters with a diameter between 100 and 200 nm (5 and 10 pixels of 20 nm), and brightness (that is, number of localizations per 20 nm pixel) > ~2-fold higher than the scattered background localizations, which most likely represent single clathrin molecules. Flat clathrin plaques were distinguished as inhomogeneous irregularly shaped clusters of localizations with brightness equal to or slightly higher than the background localizations and larger than 200 nm (10 pixels of 20 nm) in each direction. After selection, area and shape characteristics were measured using the Analyze Particles routine in FIJI.

Because pits are homogeneous in size and shape, the density of pits was calculated simply as the number of pits in the ROI divided by the ROI area. Plaques were quantified by calculating the fraction of area covered by them within the ROI, that is, the total area of all plaques in the ROI divided by the total ROI area. This measure was chosen because it includes size, shape and number of plaques, all parameters that might change considerably in our experiments.

Chromatic aberration correction: multicolour tetraspeck beads (Invitrogen) were imaged in orange (532 nm) and far-red (642 nm) channels. The resulting images and coordinates were used to obtain and correct chromatic aberration parameters using the Image Stabilizer imageJ plugin[30] (Fiji) on images, or the vec2dtransf R package on localization coordinates (for Coordinate-based Colocalization, CBC). CBC was calculated using a previously published algorithm[32] included in the ImageJ plugin Thunderstorm[62].

Tracking of CCSs from live-cell TIRF movies: time-lapse movies of internalizing Clathrin-positive structures were recorded with 1 frame per second on a Leica TIRF system, with 488 and 561 nm lasers, using a green/red filter cube and exposure time of 900 ms. Laser power was adjusted to maximize signal-to-noise ratio while limiting bleaching to < 50% over 5 min. In order to allow faithful tracking of CCSs the time lapses were preprocessed in ImageJ as described below. Movies were smoothed in time by a temporal mean filter of 3 s (3 frames) and subsequently corrected for bleaching using a custom algorithm. In brief, average fluorescence signal and background were calculated for every frame using automatic thresholding. Images were divided by the difference between signal and background and multiplied by the difference found in the first frame. Next, a

temporal median filter with a 10 s window was employed to eliminate apparent rapid intensity changes due to noise in individual CCSs. CCSs were tracked in a 3 μm-wide region bordering the periphery of the cell using the ImageJ plugin TrackMate (http://imagej.net/TrackMate). Tracking was done using the Simple LAP tracker algorithm, with a maximum travel distance of 300 nm per frame.

Local enrichment of LPAR1 in CCSs after LPA stimulation: two-colour TIRF movies of cells expressing LPAR1-GFP and CLC-RPF with time resolution 3 s and exposure time 900 ms were smoothed with a temporal mean filter using a window of 6 s (2 frames). The CLC-RFP channel was converted to a binary mask by first applying a bilateral filter (with a spatial radius of 1 pixel and intensity range 3 times the s.d. of the background), and subsequent thresholding using a semi-automatic iterative auto local threshold method. For the LPAR1-GFP channel, the median background values for each frame were determined by automatic thresholding and subsequently subtracted. The LPAR1-GFP signal, computed from CLC-RFP-positive pixels as determined by the binary mask was then normalized to the mean LPAR1-GFP in CLC-RPF signal in the frames before stimulation. Kymographs are a sequence of line profiles in time and were obtained by manually selecting an ROI from two-colour TIRF movies pre-treated as explained above and plotting the signal in this area in time, for both colours.

PIP$_3$ sensor translocation: confocal sections of cells expressing low levels of pGRP1(PH)-EGFP were acquired every 10 s. The resulting movies were subjected to image smoothing and thresholding in Fiji. To obtain membrane versus cytoplasm fluorescence tracks, a user-adjusted region was subtracted from the cell's edge and referred to as the membrane. Brightness levels of the membrane and the rest of the cell were tracked over time and the membrane/cytoplasm ratio calculated for each time point. The number of responsive cells was obtained by blind scoring of responsive cells.

**Data representation and statistical analyses.** Blots were cropped in Adobe Photoshop CS6 and uncropped scans of the most important blots are presented in Supplementary Fig. 11. R studio and GraphPad Prism (version 6.oh) were used to carry out all statistical analyses. GraphPad Prism (version 6.oh) was used to plot results always as mean ± s.e.m. of the data indicated below.

Figure 1c: $n = 15$ cells pooled from three independent experiments.

Figure 2d: $n = 15$ control KD cells, $n = 15$ N-WASP KD cells, pooled from three independent experiments.

Figure 2f: $n = 13$ control KD cells, $n = 13$ N-WASP KD cells, pooled from three independent experiments.

Figure 3c: $n = 117$ cells for GFP, $n = 54$ cells for ΔVCA, $n = 20$ cells for ΔWH1, $n = 20$ cells for ΔPRD, $n = 21$ cells for H208D, pooled from three independent experiments.

Figure 3d: $n = 12$ cells for GFP, $n = 10$ cells for ΔVCA, $n = 12$ cells for ΔWH1, $n = 10$ cells for ΔPRD, $n = 7$ cells for H208D, pooled from two independent experiments.

Figure 3e: $n = 8$ cells for GFP, $n = 10$ cells for ΔVCA, $n = 5$ cells for ΔWH1, $n = 10$ cells for ΔPRD, $n = 7$ cells for H208D, pooled from two independent experiments.

Figure 3f: $n = 12$ cells for GFP, $n = 10$ cells for ΔVCA, $n = 12$ cells for ΔWH1, $n = 10$ cells for ΔPRD, $n = 7$ cells for H208D, pooled from two independent experiments.

Figure 3g: $n = 11$ CTR KD cells, $n = 11$ ArpC2 KD #1 cells, $n = 10$ ArpC2 KD #2 cells, pooled from three independent experiments.

Figure 3j: $n = 8$ ArpC2 KD #1 + GFP cells, $n = 11$ ArpC2 KD #1 + N-WASP cells, $n = 10$ ArpC2 KD #2 + GFP cells, $n = 10$ ArpC2 KD #2 + N-WASP cells, pooled from two independent experiments.

Figure 4b: $n = 22$ control KD + 0.1% FCS cells, $n = 23$ control KD + 10% FCS cells, $n = 22$ N-WASP KD + 0.1% FCS cells, $n = 23$ N-WASP KD + 10% FCS cells, pooled from three independent experiments.

Figure 4c: $n = 19$ cells for 0, $n = 20$ cells for 3, 7, 15, 30 and 60 min, pooled from three independent experiments.

Figure 4d: $n = 5,048$ CTR KD tracks, $n = 3,619$ N-WASP KD tracks; $n = 7$ CTR KD cells, $n = 7$ N-WASP KD cells, pooled from two independent experiments.

Figure 4f: $n = 36$ cells for NS, $n = 36$ cells for EGF, $n = 36$ cells for LPA, $n = 34$ cells for FCS, pooled from three independent experiments.

Figure 4h: $n = 15$ control KD NS cells, $n = 15$ control KD + LPA cells, $n = 15$ LPAR1 KD #1 NS cells, $n = 15$ LPAR1 KD #1 + LPA cells, $n = 15$ LPAR1 KD #2 NS cells, $n = 15$ LPAR1 KD #2 + LPA cells, pooled from two independent experiments.

Figure 5e: $n = 23$ cells for DMSO, $n = 25$ cells for Dynasore, pooled from two independent experiments.

Figure 6a: $n = 21$ control KD NS cells, $n = 14$ control KD + EGF cells, $n = 21$ N-WASP KD NS cells, $n = 14$ N-WASP KD + EGF cells, pooled from three independent experiments.

Figure 6b: $n = 20$ control KD NS cells, $n = 21$ control KD + LPA cells, $n = 18$ N-WASP KD NS cells, $n = 20$ N-WASP KD + LPA cells, pooled from three independent experiments.

Figure 6c: EGF: control $n = 51$ cells for SS, $n = 51$ cells for 3 min, $n = 53$ cells for 7 min, $n = 52$ cells for 15 min, $n = 55$ cells for 15 min, $n = 54$ cells for 60 min; N-WASP KD $n = 51$ cells for SS, $n = 50$ cells for 3 min, $n = 50$ cells for 7 min, $n = 50$ cells for 15 min, $n = 53$ cells for 15 min, $n = 50$ cells for 60 min; pooled from three independent experiments, LPA: control $n = 83$ cells for SS, $n = 87$ cells for 3 min, $n = 81$ cells for 7 min, $n = 83$ cells for 15 min, $n = 87$ cells for 15 min, $n = 87$ cells for 60 min; N-WASP KD $n = 81$ cells for SS, $n = 77$ cells for 3 min, $n = 95$ cells for 7 min, $n = 70$ cells for 15 min, $n = 83$ cells for 15 min, $n = 93$ cells for 60 min; pooled from three independent experiments.

Figure 7c: $n = 12$ control KD + LPA cells, $n = 12$ control KD + EGF cells, $n = 11$ N-WASP KD + LPA cells, $n = 11$ N-WASP KD + EGF cells, pooled from three independent experiments.

Figure 7e: $n = 24$ control KD cells, $n = 15$ N-WASP KD cells, pooled from three independent experiments.

**Data availability.** All relevant data are available from the authors on reasonable request.

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

## Acknowledgements

We thank E. Boucrot (University College London) for critically reading the manuscript, and L. Lagnado (University of Sussex), D. Gadella (University of Amsterdam) and W. Moolenaar (Netherlands Cancer Institute) for sharing reagents. This work was supported by Stichting voor de Technische Wetenschappen Technology Foundation (grant number 12150 to K.J.).

## Author contributions

T.I., D.L.-P., E.A., K.J. and M.I. designed the experiments. D.L.-P., T.I., E.A., J.K. and M.I. performed the experiments. D.L.-P., T.I., E.A., B.v.d.B. and M.I. performed data analysis and quantified results. M.I. conceived and coordinated the study and K.J. supervised imaging and data analysis. M.I., D.L.-P. and K.J. wrote the manuscript.

## Additional information

**Competing interests:** The authors declare no competing financial interests.

