## [Peer Review File · Nature Communications]

Reviewers' comments:

Reviewer #1 (Remarks to the Author):

The authors present a detailed study using super-resolution microscopy (GSDIM) to study the organization and dynamics of flat clathring plaques and their involvement in clathrin-mediated endocytosis. They make some very important observations that allow interesting conclusions. This is an excellent story with impressive data and nice messages, which will have some impact in the field. Therefore, I in principle highly support publication in Nature Communications. Unfortunately, the story is hard to read – I have a few suggestions how to improve that.

- The high number of abbreviations confused me – maybe one could introduce an abbreviation list?
- Some of the data needs more quantification (e.g. co-localization, plaque size, ...) – most quantification has been done via cluster density and covered area – why is this the best way for quantification, may cluster size not be more adequate? Please justify.
- CCPs and FCPs are solely distinguished based on their sizes. As shown, both structures are highly dynamic and could easily be inter-mixed. Is there not another/additional way of identifying them (e.g. a marker or circularity and size as shown in figure 1 – or have both parameters always been used?)?
- Most data has been recorded on fixed cells – this should be written more clearly. Sorry for the usual question, but do the authors have any proof that fixation/antibody-labelling does not bias the data?
- Although done in most cases some experiments may be motivated in a better way (e.g. why FCS dependency, why live-cell recordings on page 10, why EGF on page 10, why BSC-1 cells as a justification for different cell types although it seems very special, ...).
- The authors motivate their work with the need for super-resolution microscopy. However, some of the conclusions are drawn from conventional TIRF microscopy (especially for the live-cell data) – this seems contradictory. How can they extract enough information if the resolution is not good enough?
- In some occasions (e.g. page 1 lines 248ff, delta-VCA etc on page 9ff expressed in WASP knock-out cells?) the experiment and according figures are not described/explained well enough, which makes it hard to follow the story.
- The main text is very long. The authors should consider moving some of the experiments (description + figures) to the supplements and only briefly mention them in the main text. In this way the story will be much better to follow.
- This reviewer is very cautious about possible artefacts introduced when analysing GSDIM (or STORM/PALM) images. The authors use a well-established algorithm (Thunderstorm). It would be good to see the outcome in one case (e.g. fig 1c) after using another algorithm.

Minor comments:

- The authors in some cases do not show data (e.g. page 6) – would still be good to see it (as supplementary material).
- Page 10, line 179: Increase in what – area?
- Page 12, lines 221ff: I did not understand the purpose of this paragraph – is this a short summary?
- Page 12, line 229/234: Which ligand?
- Page 13, line 262: Why figure 6b – sometimes the authors jump between figures, which makes it tough to follow the story?
- Page 16, line 16: Which nanodomains?
- Page 17, lines 321ff: I did not get this – why should F-actin not be visible in fluorescence images – too faint?

Reviewer #2 (Remarks to the Author):

General comments.

In this paper from the Innocenti lab the authors use super-resolution microscopy (SR) and antibody staining on fixed cells to image clathrin coated structures (CCS) at the plasma membrane. They conclude that there are two populations of CCS: viz individual coated pits (CCPs) and heterogeneous larger flat clathrin plaques (FCPs). The authors show that knock-down of NWASP triggers the formation of extensive FCPs. The formation of FCPs is also promoted by serum starvation and either challenge with FCS or LPA triggers dissolution of FCPs at the same time as LPAR1. Knockdown of NWASP blocks LPA triggered AKT signalling but not EGFR triggered AKT signalling. The authors conclude that FCPs are sites of clathrin mediated endocytosis which form signalling 'nano-domains' that scaffold LPAR1 mediated AKT signalling. They conclude that endocytosis of CCPs is not actin dependent but endocytosis and signalling at FCPs is actin dependent.

This paper reflects a growing interest in the field in the cell biology of clathrin plaques, structures which have hitherto been dismissed as artefacts yet which naturally form in a variety of cellular contexts. In the first half of the paper SR imaging is used to describe FCPs. However, there are some technical issues which I was unclear about. The best of the 'donut' images of mature CCPs are impressive (eg Figure 1b, middle top panel), but what about partially circular structures (eg Figure 1b, top left panel – spots and blobs) and the widely scattered small punctae? Are they poorly labelled CCPs or immature CCPs and is there a certain amount of background labelling (inevitable with antibodies coupled to sticky organic fluorophores)? How can the authors differentiate between an immature CCP and a randomly stuck antibody? Due to these issues these images are not trivial to interpret at all as while the SR images are better resolved than conventional fluorescence images they still don't come anywhere near the clarity of the original deep-etch EM images of Heuser or Anderson. So I would imagine that where (and how) the threshold and descriptors are set for pit/not pit are important. I think a panel of images from single molecule punctae through to FCP in the supplementary data would fix this just to be absolutely clear what a CCP and FCP look like, given the thresholds used by the authors, and also show objects which were discarded for one reason or another.

I am skeptical about the strength of some of the colocalization data, in particular for NWASP and FCPs (Figure 1e). The NWASP signal is in the form of a fairly uniform field of spots so some coincidence is inevitable. Likewise the colocalization between EGFR or LPAR1 and FCPs (Figure 6 a,b) seems weak to my eye. This view is reinforced by the following observation: the colocalization between FCPs and LPAR1 in the live cell imaging data is striking when compared to the SR data and I would not be surprised if fixation and processing for SR microscopy causes some receptor redistribution or loss (due to membrane extraction using detergent). This was a perennial concern back in the days before live cell imaging really took off.

The authors have been quite selective in the papers they cite and how they're cited and I have made notes on this in the detailed comments below. Suffice it to say there is published evidence, using live cell imaging and detection of single scission events using a 'pulsed pH assay' that show: i) CCPs and FCPs host discrete, quantized budding events, ii) essentially the same set of proteins are recruited to budding events hosted by CCPs and FCPs, iii) dynamin, NWASP and actin are recruited to budding events at CCPs or FCPs with the same kinetics (and NWASP recruitment is very transient) and iv) globally blocking actin polymerization using latrunculin dramatically lowered the incidence of budding events at both CCPs and FCPs in the cells used (NIH-3T3). These published results are relevant here and should be discussed.

The finding that NWASP knockdown augments FCP formation is new and interesting. The authors should also discuss the links between plaques and integrin adhesion and possible impact of NWASP knock down on global cell mechanics (ie membrane tension / adhesion / whether the cell is still

crawling or not).

The finding that NWASP knockdown augments LPA mediated AKT signalling is surprising and interesting (Figure 7b) but it's also really surprising that LPA doesn't stimulate AKT in the control cells at all - given that the dogma is that LPA stimulation normally results in robust signalling from PI3K and resultant AKT phosphorylation on Thr308 and Ser473. I think that this result should be double checked and confirmed. One prediction from the authors central thesis is that LPA mediated AKT signalling should be sensitive to general perturbation of clathrin - so this is something the authors should test. Also, did the authors look to see if phosphor-AKT is localized to FCPs? That would be useful information.

So, to conclude, the first part is incremental and doesn't add much new to the debate on plaques (and the authors should cite previous papers - no point reinventing the wheel). The finding that NWASP knock down augments the formation of FCPs is new and interesting and the links between FCPs and adhesion should be discussed (ie why do FCPs form in the first place? this seems very relevant here). The link between LPA/LPAR1, NWASP and AKT signaling is new and interesting and should be developed further and additional efforts made to really pin it to FCPs.

Detailed notes:

1. Line 47. Should also cite 'N-WASP deficiency impairs EGF internalization and actin assembly at clathrin-coated pits. Benesch S, Polo S, Lai FP, Anderson KI, Stradal TE, Wehland J, Rottner K. J Cell Sci. 2005 Jul 15;118(Pt 14):3103-15.
2. Line 69. Along with 11,19,23,24 should also cite ref. 12. There is a lively debate whether clathrin plaques are sites of active CME. Kirchhausen claimed that they represent a 'non-classical' site of endocytosis that form 'mesa-like' structures (Saffarian & Kirchhausen 2009, ref. 13 here) which are quite unlike clathrin coated vesicles (... so ref. 13 is inappropriate here - they absolutely DID NOT claim that plaques are sites of regular CME!). On the other hand Heuser showed clathrin buds at the edge of plaques (ref. 18 here) and Pollard showed budding domains within plaques (ref. 19 here) and the Merrifield lab has repeatedly shown that clathrin plaques are the sites of discrete clathrin mediated endocytic scission events (refs. 11, 12 and for GPCRs see Lampe & Merrifield, 2014).
3. Lines 72-81. It was shown previously using live cell imaging and an assay to detect single scission events that both pits and plaques are endocytically active and host 'regular' scission events - which recruit essentially the same suite of proteins including dynamin, NWASP and actin, at least in NIH-3T3 and HEK293 cells (Taylor & Merrifield, 2011; ref 12 here also Lampe & Merrifield 2014).
4. The best of the SR data is really impressive. However, to make a donut the pit has to be quite mature - so what are all the other little spots and blobs? Are they immature coated pits or non-specific labelling (which will be inevitable - you can never get zero background with antibodies)? Given this uncertainty how do the authors set their cut-off? A panel in supplementary data showing structures from isolated diffraction limited spots through to FCPs and making the cut-off clear would be helpful here.
5. The colocalization between FCPs and F-actin is a bit sketchy and there are FCPs which don't colocalize with any F-actin at all (upper panels). The colocalization between FCPs and NWASP is really weak - there is a lot of uniform punctate NWASP label. Not surprising given NWASP is so transiently recruited.
6. The increase in FCPs on NWASP knock down is new and intriguing. However, given that FCPs are sites of adhesion can't help wondering what this means in terms of global cell mechanics (are the cells better stuck down? can they still crawl?).

6. Dissolution of FCPs on addition of FCS or LPA looks interesting (Figure 4) but I would like to see a whole cell movie for Figure 4d.

7. Figure 7b. This seems the most novel and striking result of all. But does LPAR1 mediate the dissolution of FCPs triggered by FCS or LPA (evidence)? The authors need to do a LPAR1 knockdown here. They should also use a general clathrin blocker (Pitstop2?) to show whether FCS/LPA stimulated AKT signalling is dependent on clathrin turnover and also use a general actin blocker (eg latrunculin-B) to strengthen 7b. The finding that AKT is NOT activated at all in the control cells seems very (very) surprising. That needs to be carefully checked.

General

We are grateful for a set of excellent reviews. Both Reviewers have raised a number of relevant points that we have duly addressed in the accompanying revised manuscript.

We have changed the text significantly at several locations, touched up textual details elsewhere and added all requested additional data. This necessitated substantial revision of the figures and addition of six new supplemental figures. We also included two figures for Reviewers only to help them assessing our claims.

Details will be given below:

- Reviewer #1 (page 2-5)
- Reviewer #2 (page 6-14)

Reviewer #1

The authors present a detailed study using super-resolution microscopy (GSDIM) to study the organization and dynamics of flat clathrin plaques and their involvement in clathrin-mediated endocytosis. They make some very important observations that allow interesting conclusions. This is an excellent story with impressive data and nice messages, which will have some impact in the field. Therefore, I in principle highly support publication in Nature Communications. Unfortunately, the story is hard to read – I have a few suggestions how to improve that.

R: We thank the reviewer for supporting the publication of our study. We have done our best to improve readability by editing the text.

- The high number of abbreviations confused me – maybe one could introduce an abbreviation list?

R: As suggested by the Reviewer, we have added an abbreviation list in the new Supplementary Fig. S1c. We have also cut back on the use of abbreviations, e.g. by substituting ‘pits’ and ‘plaques’ for CCPs and FCPs, respectively, after clearly defining those terms in the context of this paper.

- Some of the data needs more quantification (e.g. co-localization, plaque size, ...) – most quantification has been done via cluster density and covered area - why is this the best way for quantification, may cluster size not be more adequate? Please justify.

R: The figures show that different experimental conditions alter both the number and size of plaques considerably; that is why we think that the total covered area is a fair measure for plaques, since it includes both shape and number. On the other hand, pits are much more homogenous in size and shape and therefore a simple count represents a fair measure.

We agree with the Reviewer that, without justification, these criteria seemed arbitrary and have now added this information to the Methods section.

- CCPs and FCPs are solely distinguished based on their sizes. As shown, both structures are highly dynamic and could easily be inter-mixed. Is there not another/additional way of identifying them (e.g. a marker or circularity and size as shown in figure 1 – or have both parameters always been used?)?

R: Unfortunately, we are not aware of any molecular markers that would reliably distinguish between the two clathrin-coated structures. Small size and circularity are the criteria that set pits apart from plaques. Please note that we in fact did combine size as well as circularity of the pits, as we have now emphasized more clearly in the manuscript. Automated analysis is not always reliable for circularity (e.g. when two pits touch each other) and therefore all individual pits and plaques have been judged by eye (computer-assisted analysis, rather than fully automated analysis). The Methods section has also been

touched up to better clarify this. The new Supplemental Figure 1 should help the readers judge the validity of this approach.

- Most data has been recorded on fixed cells – this should be written more clearly. Sorry for the usual question, but do the authors have any proof that fixation/antibody-labelling does not bias the data?

R: For the sake of compactness, we had left this information largely out of the manuscript. At the onset of this project, we had extensively compared and optimized fixation and labeling, and we feel confident that our optimized fixation and labeling protocols preserve details as much as possible. This work was recently published separately (Leyton-Puig et al., Biol Open 2016). We have now included supporting data that reiterate the validity of the results presented in that paper, except that here the focus is on clathrin structures. We added text (Results section; Methods section) and Supplementary Figure 9, which summarizes results obtained with different fixation methods and different analysis algorithms.

We also note that the excellent colocalization between antibodies against clathrin heavy chain and against GFP (for GFP-tagged clathrin light chain; see Supplementary Fig. S2a) rules out a bias generated by these antibodies.

- Although done in most cases some experiments may be motivated in a better way (e.g. why FCS dependency, why live-cell recordings on page 10, why EGF on page 10, why BSC-1 cells as a justification for different cell types although it seems very special, ...).

R: We thank the Reviewer for this useful remark. To increase the readability of the text, we have explained more clearly both the rationales and the experiments in question.

- The authors motivate their work with the need for super-resolution microscopy. However, some of the conclusions are drawn from conventional TIRF microscopy (especially for the live-cell data) – this seems contradictory. How can they extract enough information if the resolution is not good enough?

R: It proved not feasible to do live-cell time-lapse SR by GS-DIM/STORM in our cells. It is true that TIRF only provides high resolution in axial direction, while its lateral resolution is not high enough to reliably discriminate plaques from clusters of pits. For that reason, we also tried structured illumination microscopy (SIM; up to 2-fold increase in lateral resolution) and re-scan confocal microscopy (up to 1.5 times increased lateral resolution). However, both techniques have Z resolution much worse than TIRF and therefore yielded results that were impossible to interpret.

We therefore had to rely on TIRF for the time-lapse experiments (Fig. 4d; Fig. 5a and Fig. 7d-f). Still, we uphold that our time-lapse results are reliable because in all cases we also used SR to characterize parallel samples, fixed at selected time points. This and the excellent correlation between TIRF data and SR data allow us interpreting what we see in TIRF. Of course, without our SR data it would have been impossible to interpret the time-lapse TIRF data.

- In some occasions (e.g. page 1 lines 248ff, delta-VCA etc on page 9ff expressed in WASP knock-out cells?) the experiment and according figures are not described/explained well enough, which makes it hard to follow the story.

R: We thank the Reviewer again for helping us to improve the text.

- The main text is very long. The authors should consider moving some of the experiments (description + figures) to the supplements and only briefly mention them in the main text. In this way the story will be much better to follow.

R: We have streamlined the text as much as possible by cutting all unnecessary and redundant text in all sections of the revised manuscript.

- This reviewer is very cautious about possible artefacts introduced when analysing GSDIM (or STORM/PALM) images. The authors use a well-established algorithm (Thunderstorm). It would be good to see the outcome in one case (e.g. fig 1c) after using another algorithm.

R: This is an important point that we have addressed in Supplementary Fig. 9c, which demonstrates that leading analysis algorithms like Thunderstorm, rapidSTORM, quickPALM and the built-in Leica algorithm (LAS AF) all give comparable final images.

Minor comments:

- The authors in some cases do not show data (e.g. page 6) – would still be good to see it (as supplementary material).

R: The EM data have been added as requested (see Supplementary Fig.2b).

- Page 10, line 179: Increase in what – area?

R: We meant the increase in the plaque-covered area. We have touched up the text.

- Page 12, lines 221ff: I did not understand the purpose of this paragraph – is this a short summary?

R: This paragraph has been removed.

- Page 12, line 229/234: Which ligand?

R: LPA and EGF for the LPAR1 and the EGFR, respectively. The text has been amended.

- Page 13, line 262: Why figure 6b – sometimes the authors jump between figures, which makes it tough to follow the story?

R: We have noticed and amended these inconsistencies.

- Page 16, line 16: *Which nanodomains?*

R: The plaques. We amended the text.

- Page 17, lines 321ff: *I did not get this – why should F-actin not be visible in fluorescence images – too faint?*

R: This is actually the most likely explanation. Faint actin staining may be hard to see when the other color channel is bright.

Reviewer #2

General comments.

In this paper from the Innocenti lab the authors use super-resolution microscopy (SR) and antibody staining on fixed cells to image clathrin coated structures (CCS) at the plasma membrane. They conclude that there are two populations of CCS: viz individual coated pits (CCPs) and heterogeneous larger flat clathrin plaques (FCPs). The authors show that knock-down of NWASP triggers the formation of extensive FCPs. The formation of FCPs is also promoted by serum starvation and either challenge with FCS or LPA triggers dissolution of FCPs at the same time as LPAR1. Knockdown of NWASP blocks LPA triggered AKT signalling but not EGFR triggered AKT signalling. The authors conclude that FCPs are sites of clathrin mediated endocytosis which form signalling 'nano-domains' that scaffold LPAR1 mediated AKT signalling. They conclude that endocytosis of CCPs is not actin dependent but endocytosis and signalling at FCPs is actin dependent.

This paper reflects a growing interest in the field in the cell biology of clathrin plaques, structures which have hitherto been dismissed as artefacts yet which naturally form in a variety of cellular contexts.

R: We were delighted to read that this Reviewer finds that clathrin plaques are a hot topic in cell biology.

In the first half of the paper SR imaging is used to describe FCPs. However, there are some technical issues which I was unclear about. The best of the 'donut' images of mature CCPs are impressive (eg Figure 1b, middle top panel), but what about partially circular structures (eg Figure 1b, top left panel – spots and blobs) and the widely scattered small punctae? Are they poorly labelled CCPs or immature CCPs and is there a certain amount of background labelling (inevitable with antibodies coupled to sticky organic fluorophores)? How can the authors differentiate between an immature CCP and a randomly stuck antibody?

Due to these issues these images are not trivial to interpret at all as while the SR images are better resolved than conventional fluorescence images they still don't come anywhere near the clarity of the original deep-etch EM images of Heuser or Anderson. So I would imagine that where (and how) the threshold and descriptors are set for pit/not pit are important.

I think a panel of images from single molecule punctae through to FCP in the supplementary data would fix this just to be absolutely clear what a CCP and FCP look like, given the thresholds used by the authors, and also show objects which were discarded for one reason or another.

R: We thank the Reviewer for appreciating the quality of our images, and have tried to address his concerns in several ways.

We have added a representative image showing cells stained only with the secondary antibody (each conjugated to 4-5 dye molecules), which results in a slight background labeling (Supplementary Fig. S9b). It can be seen that the intensity of such

single antibody molecules is much less than that of the pits because each pit is labeled by approximately 20 antibody molecules. On top of that, the diameter of single Ab in the SR images averages ~ 3 pixels, whereas pits typically span 7-11 pixels. Thus, we conclude that most of the partially circular structures are immature CCPs. As even single clathrin molecules may be visible at this resolution, we cannot formally distinguish whether the scattered punctae are background or membrane-bound triskelia. We have briefly clarified this issue in the revised text, and added a more complete description to the Methods section where we explain the classification criteria.

Furthermore, we made a gallery of CHC-positive objects, classified as either pits or plaques, that illustrates our classification criteria as requested by the Reviewer (Supplementary Fig. 1a-b).

Finally, we would like to point out that although the quality of SR images does not match that of deep-etch EM work, only SR light microscopy enables collecting the large numbers of images necessary for robust statistical analysis of many cell lines and conditions, and combining these with TIRF overview images. We feel that an occasional miss-classification should not alter the message since our results are highly significant.

I am skeptical about the strength of some of the colocalization data, in particular for NWASP and FCPs (Figure 1e). The NWASP signal is in the form of a fairly uniform field of spots so some coincidence is inevitable.

R: There is no doubt that NWASP is present in clathrin-positive structures as previously shown by us (Innocenti et al., NCB 2005; Galovic et al., JCS 2011) and the Merrifield lab (Taylor et al., PLoS Biol 2011). Coordinate-based colocalization coefficient (CBC) for full-length N-WASP and plaques was reported in Figure 3d. Those quantitative analyses indicate that N-WASP is enriched (even if relatively modestly) in plaques when compared to both GFP alone and some N-WASP mutants. We did not intend to suggest that they are exclusively present together. We have edited the first paragraph of the Results section to reflect this point more clearly.

The concept of colocalization in SR is distinct from that in wide-field fluorescence microscopy. With increasing resolution, colors overlap less and less, and therefore CBC had to be developed. Unfortunately, our eyes are used to seeing colocalization as ‘orange overlap’ and it is therefore much harder to visually capture colocalization in SR images. We have also replaced the image depicting N-WASP false colored in red with a new one that shows N-WASP in green thereby allowing human eye to better see the contrast (Fig. 1e). Also, see our answer to the next point.

Likewise the colocalization between EGFR or LPAR1 and FCPs (Figure 6 a,b) seems weak to my eye. This view is reinforced by the following observation: the colocalization between FCPs and LPAR1 in the live cell imaging data is striking when compared to the SR data and I would not be surprised if fixation and processing for SR microscopy causes some receptor redistribution or loss (due to membrane extraction using detergent). This was a perennial concern back in the days before live cell imaging really took off.

R: This is an important issue; we apologize that our response here is lengthy.

With respect to the difference between SRm and TIRFm, it is important to realize

that they represent quite different imaging modalities. In TIRFm, the intensity of pixels is determined (apart from microscope parameters) by the number of fluorophores present, as well as by the proximity to the glass coverslip. Close to the coverslip the intensity is actually several-fold higher than in epi-fluorescence (Martin-Fernandez et al, J. Microscopy 2013), dropping exponentially towards zero at a distance of ~ 150 nm. Therefore structures that are closer to the coverslip have exaggerated brightness, when compared to normal wide-field fluorescence microscopy. In SRm, fluorophore localizations (as opposed to intensities) are depicted: each gray level in a pixel represents one localization (or blink), irrespective of its intensity (this is equally true whether SRm is acquired in TIRF mode or in epi mode). The threshold for detection of blinking is set just above noise level to retrieve as much localizations as possible. As a consequence of this, the selective exaggeration of intensity seen in TIRF is partly lost in SR images. On top of that, the dotted appearance of SR images makes it harder to judge colocalization: indeed, as pixels become smaller, exact colocalization becomes very rare. Therefore, CBC (see previous point) actually reports molecular proximity and not exact colocalization. These considerations make it clear why only quantitative analysis of SR images allows drawing firm conclusions, and such analysis (Figure 6) fully supports all our claims.

To illustrate the above concepts, in Figure 1 for Reviewers we compared SR data for LPAR1 and clathrin heavy chain (right panels correspond to the dashed boxed area) prior (top row) and after applying a Gaussian smoothing filter of 250 nm radius to mimic conventional microscope resolution (bottom row). It can be seen that the prevalence of yellow overlap visible in the bottom panels is hard to observe in the SR images.

Leyton-Puig et. al. Figure 1 for Reviewers

Nevertheless, to further alleviate the concern of the Reviewer, we have now also quantified the colocalization of EGFR and LPAR1 in a large non-plaque membrane region of the cells. These data show much less colocalization and have been added as new bars to Figure 6a and 6b.

Finally, we agree with the Reviewer that fixation may theoretically perturb protein retrieval, distribution and visualization. However, we have taken every precaution to prevent this as much as possible, as is now also better emphasized in the manuscript.

The authors have been quite selective in the papers they cite and how they're cited and I have made notes on this in the detailed comments below. Suffice it to say there is published evidence, using live cell imaging and detection of single scission events using a 'pulsed pH assay' that show: i) CCPs and FCPs host discrete, quantized budding events, ii) essentially the same set of proteins are recruited to budding events hosted by CCPs and FCPs, iii) dynamin, NWASP and actin are recruited to budding events at CCPs or FCPs with the same kinetics (and NWASP recruitment is very transient) and iv) globally blocking actin polymerization using latrunculin dramatically lowered the incidence of budding events at both CCPs and FCPs in the cells used (NIH-3T3). These published results are relevant here and should be discussed.

R: We have added the references in question and discussed this literature in the context of our findings where necessary.

The finding that NWASP knockdown augments FCP formation is new and interesting. The authors should also discuss the links between plaques and integrin adhesion and possible impact of NWASP knock down on global cell mechanics (ie membrane tension / adhesion / whether the cell is still crawling or not).

R: We thank the Reviewer for these useful suggestions.

We have extended the characterization of FCPs by including integrin $\beta 5$ (Supplementary Fig. S2f), which has been previously shown to localize to FCPs, and integrin $\beta 1$ (not shown). The lack of integrin $\beta 1$ enrichment and absence of stress fibers shows that these plaques are functionally distinct from focal adhesions. In keeping with this, the control KD and the N-WASP KD cells showed similar integrin-dependent and integrin-independent adhesive properties (Supplementary Fig. S8b). Yet, N-WASP KD cells migrate less than control cells both in serum-starved and EGF-stimulated conditions (Supplementary Fig. S8a). Importantly, EGF does not perturb FCPs, and both EGF-induced activation of AKT (this study) as well as membrane ruffling (Innocenti et al., NCB 2005) are independent of N-WASP. Therefore, these results suggest that FCPs oppose cell motility by fulfilling anti-migratory structural and/or signaling roles. Thus, FCPs appears to be atypical sites of adhesion from a functional viewpoint.

We have also explored the link between FCPs and membrane tension. In our hands, a reduction in tension induced by either global actin depolymerization or ROCK inhibition does not translate into a reduction in FCPs. Our results support the emerging view that FCP presence does not depend on tension. These results have been added to the manuscript.

The finding that NWASP knockdown augments LPA mediated AKT signalling is surprising and interesting (Figure 7b) but it's also really surprising that LPA doesn't stimulate AKT in the control cells at all - given that the dogma is that LPA stimulation normally results in robust signalling from PI3K and resultant AKT phosphorylation on Thr308 and Ser473. I think that this result should be double checked and confirmed.

R: We respectfully point out that LPA does activate AKT in the control KD cells, it is just less effective in doing that. To ease the Reviewer's mind, below is a longer exposure of the blot depicted in Fig. 7b:

One prediction from the authors central thesis is that LPA mediated AKT signalling should be sensitive to general perturbation of clathrin – so this is something the authors should test.

R: We used the inhibitor pitstop2 to freeze the turnover of clathrin as suggested by the Reviewer in her/his detailed notes.

Given that mounting evidence casts serious doubts about the claimed specificity of this inhibitor (*e.g.* Willox et al., Bio Open 2014), we decided to characterize its effects on FCPs and CCPs.

As the Reviewer can see below (Figure 2 for Reviewers), HeLa cells treated with 30 μ M PitStop2 for 15 minutes prior to fixation and staining underwent an unexpected reduction in FCPs, whereas CCPs were not affected. At the biochemical level, PitStop2 decreased the activity of AKT in LPA stimulated cells, regardless of N-WASP expression levels, which would fit with our hypothesis that plaques function as LPAR1 signaling domains. However, PitStop2 also led to the constitutive activation of ERK in both control and N-WASP KD cells. Moreover, similar observations were made using EGF (not shown), thereby demonstrating that these effects are independent of the employed agonist. We also note that upon pretreatment with pitstop2 cells looked really deplorable. Given that PitStop2 perturbs cell signaling in a way that it is independent of both N-WASP (and ensuing FCP dynamics) and extracellular stimuli, we prefer to exclude these results from the manuscript, because we believe that its mechanism of action is currently unclear.

To further corroborate the link between AKT activity and FCPs, we have taken advantage of the Arpc2 KD cells, which phenocopy the increase in FCPs found in the N-WASP KD cells (Fig. 3g-j). In good agreement with these data, we discovered that hyperactivation of AKT by LPA also occurs upon knockdown of the Arp2/3 complex. These new results can be seen in Fig. 7g.

Leyton-Puig et. al. Figure 2 for Reviewers

Also, did the authors look to see if phosphor-AKT is localized to FCPs? That would be useful information.

R: We looked at the localization of phosphorylated AKT using both anti-pSer473 and anti-pThr308 antibodies. The former antibodies gave a non-specific signal that did not increase after LPA stimulation. Conversely, the latter ones resulted in an LPA-inducible, yet diffused membrane pattern (not shown). Based on these results, it seems that activated AKT is not confined within FCPs.

So, to conclude, the first part is incremental and doesn't add much new to the debate on plaques (and the authors should cite previous papers – no point reinventing the wheel). The finding that NWASP knock down augments the formation of FCPs is new and interesting and the links between FCPs and adhesion should be discussed (ie why do

FCPs form in the first place? this seems very relevant here). The link between LPA/LPAR1, NWASP and AKT signaling is new and interesting and should be developed further and additional efforts made to really pin it to FCPs.

R: We would like to emphasize here that the first part goes beyond a mere incremental advance as it reports that N-WASP localizes at FCPs and plays a crucial role in the regulation thereof.

Most importantly, we have explored the regulation and function of FCPs as explained in detail both above and below. In short, we found that *i)* FCP abundance correlates inversely with cell migration but has no role in either integrin-dependent or integrin-independent adhesion (Supplemental Fig. 8), *ii)* FCP presence is not strictly dependent on high tension (Supplemental Fig. 7), *iii)* FCP dissolution induced by LPA functionally requires LPAR1 (Fig. 4) and *iv)* LPA-dependent regulation of AKT signaling through FCP dissolution involves not only N-WASP but also the Arp2/3 complex (Fig. 7).

In summary, we believe that these new results further illuminate FCP regulation and function.

Detailed notes:

1. Line 47. Should also cite 'N-WASP deficiency impairs EGF internalization and actin assembly at clathrin-coated pits. Benesch S, Polo S, Lai FP, Anderson KI, Stradal TE, Wehland J, Rottner K. J Cell Sci. 2005 Jul 15;118(Pt 14):3103-15.

R: We did not cite this paper because it exploits a different cell line to show the same results that Innocenti and colleagues have published in the main and the supplemental figures (NCB 2005).

2. Line 69. Along with 11,19,23,24 should also cite ref. 12. There is a lively debate whether clathrin plaques are sites of active CME. Kirchhausen claimed that they represent a 'non-classical' site of endocytosis that form 'mesa-like' structures (Saffarian & Kirchhausen 2009, ref. 13 here) which are quite unlike clathrin coated vesicles (... so ref. 13 is inappropriate here – they absolutely DID NOT claim that plaques are sites of regular CME!). On the other hand Heuser showed clathrin buds at the edge of plaques (ref. 18 here) and Pollard showed budding domains within plaques (ref. 19 here) and the Merrifield lab has repeatedly shown that clathrin plaques are the sites of discrete clathrin mediated endocytic scission events (refs. 11, 12 and for GPCRs see Lampe & Merrifield, 2014).

R: We thank the Reviewer for pointing out this mistake that originated from an incorrect and hasty reshuffling of key references. This has been amended in the revised text.

3. Lines 72-81. It was shown previously using live cell imaging and an assay to detect single scission events that both pits and plaques are endocytically active and host 'regular' scission events – which recruit essentially the same suite of proteins including dynamin, NWASP and actin, at least in NIH-3T3 and HEK293 cells (Taylor & Merrifield, 2011; ref 12 here also Lampe & Merrifield 2014).

R: This point is well taken. We have added and briefly discussed these studies.

4. The best of the SR data is really impressive. However, to make a donut the pit has to be quite mature – so what are all the other little spots and blobs? Are they immature coated pits or non-specific labelling (which will be inevitable – you can never get zero background with antibodies)? Given this uncertainty how do the authors set their cut-off? A panel in supplementary data showing structures from isolated diffraction limited spots through to FCPs and making the cut-off clear would be helpful here.

R: Agreed. This point has been answered before. We have provided a panel of structures (Supplementary Fig. 1a-b) to show the different shapes of CCPs and FCPs, and indicated what we consider to be background labeling and/or very immature pits. In the Methods section, we have also more clearly indicated the criteria used to discriminate pits from plaques and background labeling. Supplementary Figure 9b shows a representative SR image obtained when cells were labeled only with secondary antibody.

5. The colocalization between FCPs and F-actin is a bit sketchy and there are FCPs which don't colocalize with any F-actin at all (upper panels). The colocalization between FCPs and NWASP is really weak – there is a lot of uniform punctate NWASP label. Not surprising given NWASP is so transiently recruited.

R: This is an extension of one of the Reviewer's general comments. In addition to what we have already written above, we would like to stress here that the transient presence of both N-WASP and F-actin at FCPs likely explains the relatively modest "colocalization" with CHC (Fig. 3d).

F-actin is not abundant in FCPs as previous EM studies have demonstrated. Moreover, actin polymerization bursts at sites of CCV formation only after the arrival of N-WASP and just before vesicle pinching off. This explains why actin-negative FCPs could be observed in Figure 1c. This point has now been shortly mentioned in the revised text.

6. The increase in FCPs on NWASP knock down is new and intriguing. However, given that FCPs are sites of adhesion can't help wondering what this means in terms of global cell mechanics (are the cells better stuck down? can they still crawl?).

R: As explained in great detail in our response to the Reviewer's general comments, FCPs do not impact on either integrin-dependent or integrin-independent adhesion. Yet, FCPs *per se* oppose cell motility by fulfilling anti-migratory structural and/or signaling roles. Thus, FCPs appears to be atypical sites of adhesion from a functional viewpoint.

6. Dissolution of FCPs on addition of FCS or LPA looks interesting (Figure 4) but I would like to see a whole cell movie for Figure 4d.

R: Agreed (Supplementary Movie S1).

7. Figure 7b. This seems the most novel and striking result of all. But does LPAR1

mediate the dissolution of FCPs triggered by FCS or LPA (evidence)? The authors need to do a LPAR1 knockdown here.

R: We have knocked down LPAR1 in HeLa cells as requested by the Reviewer. We found that FCPs in resting conditions are not affected by the silencing of the LPAR1 (Fig. 4g-h). In line with our previous results and working model, FCPs did not dissolve in the LPAR1 KD cells upon LPA stimulation (Fig. 4g-h). Thus, we conclude that activated LPAR1 controls FCP dynamics.

They should also use a general clathrin blocker (Pitstop2?) to show whether FCS/LPA stimulated AKT signalling is dependent on clathrin turnover and also use a general actin blocker (eg latrunculin-B) to strengthen 7b.

R: Our results with pitstop2 have been discussed above.

In our opinion, the Arpc2 KD cells provide a more selective way to interfere with FCPs than Latrunculin. Thus, we challenged these cells and control cells with LPA and found that downregulation of the Arp2/3 complex leads to AKT hyperactivation (Fig. 7g).

Nevertheless, we have also employed Latrunculin A to further assess the link between actin dynamics, FCPs and signaling. As expected and already reported by other groups, Latrunculin A caused a decreased in CCPs. In line with our observations and data from the Merrifield lab, we recorded a significant increase in FCPs in Latrunculin A-treated cells (Supplementary Fig. S7a). However, Latrunculin A dampened AKT activation triggered by either LPA or EGF stimulation in both control and N-WASP KD cells (Supplementary Fig. S7b). Hence, optimal AKT activation by both LPA and EGF requires an additional N-WASP-independent, yet actin-sensitive event that occurs upstream of or parallel to FCP regulation. In this regard, previous studies have reported a number of unexpected and cell-type-dependent effects caused by global actin depolymerization both on EGFR dimerization, mobility and signaling (Chung et al, Nature 2010; Needham et al, Nat. Commun. 2016)) and LPA-induced signaling (Yu et al., Cell 2012; Sotiropoulos et al., Cell 1999).

The finding that AKT is NOT activated at all in the control cells seems very (very) surprising. That needs to be carefully checked.

R: As explained in our response to the Reviewer's general comments, AKT activation by LPA does occur in the control KD cells. Furthermore, we now show that LPA hyperactivates AKT in both the N-WASP KD cells and the Arpc2 KD cells (Fig. 7g).

REVIEWERS' COMMENTS:

Reviewer #1 (Remarks to the Author):

The authors have well commented to all of my original concerns and have revised the manuscript accordingly. I support publication in Nature Communications as is.

Reviewer #2 (Remarks to the Author):

The authors have done a very thorough job of answering my comments (and the comments of the other reviewers) and I think this paper is going to draw a lot of attention in the field.

I only have one further comment and that is that the authors should double check their references are correctly numbered (eg line 117, I think ref 12 should be ref 11). There may be other instances so a careful proof read is in order.

To conclude I would like to take this opportunity to offer my congratulations on a very interesting and provocative paper.

General

We would like to thank both Reviewers for their constructive criticisms, which have allowed us to improve our study. We really enjoyed the peer-review process.

Reviewer #1

The authors have well commented to all of my original concerns and have revised the manuscript accordingly. I support publication in Nature Communications as is.

R: We thank the Reviewer for her/his continuous support and very useful comments.

Reviewer #2

The authors have done a very thorough job of answering my comments (and the comments of the other reviewers) and I think this paper is going to draw a lot of attention in the field.

R: Many thanks to the Reviewer for these enthusiastic words.

I only have one further comment and that is that the authors should double check their references are correctly numbered (eg line 117, I think ref 12 should be ref 11). There may be other instances so a careful proof read is in order.

R: The Reviewer is right. We also realized that bug in our Endnote resulted in irrelevant references being inserted here and there. These mistakes have been corrected in the revised text.

To conclude I would like to take this opportunity to offer my congratulations on a very interesting and provocative paper.

R: Thanks again.